# Biomaterial-Based Responsive Nanomedicines for Targeting Solid Tumor Microenvironments

**DOI:** 10.3390/pharmaceutics16020179

**Published:** 2024-01-26

**Authors:** Konstantinos Avgoustakis, Athina Angelopoulou

**Affiliations:** 1Department of Pharmacy, School of Health Sciences, University of Patras, 26504 Patras, Greece; avgoust@upatras.gr; 2Clinical Studies Unit, Biomedical Research Foundation Academy of Athens (BRFAA), 4 Soranou Ephessiou Street, 11527 Athens, Greece; 3Department of Chemical Engineering, Polytechnic School, University of Patras, 26504 Patras, Greece

**Keywords:** biomaterials, tumor microenvironment, nanomedicine, hypoxia, acidosis, resistance, tumor vasculature, targeting, stimuli-responsiveness

## Abstract

Solid tumors are composed of a highly complex and heterogenic microenvironment, with increasing metabolic status. This environment plays a crucial role in the clinical therapeutic outcome of conventional treatments and innovative antitumor nanomedicines. Scientists have devoted great efforts to conquering the challenges of the tumor microenvironment (TME), in respect of effective drug accumulation and activity at the tumor site. The main focus is to overcome the obstacles of abnormal vasculature, dense stroma, extracellular matrix, hypoxia, and pH gradient acidosis. In this endeavor, nanomedicines that are targeting distinct features of TME have flourished; these aim to increase site specificity and achieve deep tumor penetration. Recently, research efforts have focused on the immune reprograming of TME in order to promote suppression of cancer stem cells and prevention of metastasis. Thereby, several nanomedicine therapeutics which have shown promise in preclinical studies have entered clinical trials or are already in clinical practice. Various novel strategies were employed in preclinical studies and clinical trials. Among them, nanomedicines based on biomaterials show great promise in improving the therapeutic efficacy, reducing side effects, and promoting synergistic activity for TME responsive targeting. In this review, we focused on the targeting mechanisms of nanomedicines in response to the microenvironment of solid tumors. We describe responsive nanomedicines which take advantage of biomaterials’ properties to exploit the features of TME or overcome the obstacles posed by TME. The development of such systems has significantly advanced the application of biomaterials in combinational therapies and in immunotherapies for improved anticancer effectiveness.

## 1. Introduction

Cancer incidence and mortality has increased dramatically, with female breast cancer being the most commonly diagnosed, surpassing lung cancer [1,2]. It represents a major public health issue in emerging and developing countries, and poses great socioeconomic and psychological challenges. According to recent world statistics, cancer is the first or second leading cause of death; there are nearly 20 million new cases and almost 10 million deaths [1]. Global cancer statistics have estimated that near to 30 million new cases should be expected by 2040, and this is expected to affect more developed than emerging economies. Due to migration and demographic changes, the rate of cancer incidence is seriously affected by everyday risk factors, such as tobacco and alcohol use, unhealthy diet, and anxiety [1,2]. Despite the disappointing statistics, however, there was a decline in the overall cancer death rate of about 33% between 1991 and 2020. Furthermore, an estimated 4 million deaths were prevented. Moreover, a decline of 65% in cervical cancer incidence among women in their 20s during the period 2012 to 2019 was observed. This is due to the preventive effect of the human papillomavirus vaccine [1,2]. The decline in rates of cancer mortality can, undoubtedly, be related to increased research efforts in the field of cancer vaccines (RNA technologies) and personalized nanomedicines [3]. The greatest challenges for scientists are to ensure early diagnosis and prevention of cancer, which could effectively reduce cancer mortality.

Early diagnosis is crucial due to the high differentiation rate of tumor cells that promote the development of highly aggressive cancerous cells associated with multidrug resistance (MDR), stemness [4], and invasion [5]. A critical stimulus of MDR and invasion is tumor microenvironment (TME) (Figure 1). This is a complicated interpenetrating network of varied cancerous and stromal cell types, extracellular matrix (ECM), and interstitial fluid (IF) [6,7,8]. TME is hostile to normal cells while being hospitable to stromal cells that are the nonmalignant components of solid tumors. These include endothelial cells (ECs), fibroblasts (FCs), immune cells (lymphocytes, macrophages, dendritic cells), and perivascular cells (PCs). These cells are interconnected in a protein-rich matrix that promotes angiogenesis and neovascularization [9,10]. Within the heterogenic TME, vasculature abnormalities are related to variations in oxygenation. Furthermore, the elevated presence of reactive oxygen species (ROS), glutathione (GSH), enzymes, and adenosine triphosphate (ATP) further promote a hypoxic status with acidic pH levels (pH 5.5–6.2). These features—in combination with secreted growth factors, cytokines, chemokines, and macromolecules such as proteases and proteins in the surrounding stroma—regulate the stimulation of cancer-associated fibroblasts (CAFs), and play key role in metastatic potency [11,12,13]. The stroma in combination with the highly dynamic ECM act as supportive reservoirs that directly or indirectly interconnect TME with capillary and vascular system cells, and immune system cells. This provide the essential nutrition components of oxygen, gas exchange, and metabolites withdrawal, to support tumorigenesis and continuous neovascularization [14,15,16].

The greatest difficulty posed by the TME of solid tumors is MDR. This results in reduced therapeutic efficiency of traditional interventions such as chemo- and radiotherapy. The backbone of traditional therapeutic approaches is surgical ablation followed by chemo- and radiotherapy or a combination of both, depending on tumor severity. Chemo- and radiotherapy cause serious side effects for the patients within the therapeutic window of the administered doses [17]. Great progress has been achieved with advanced investigation of new therapeutic agents including peptides, antibodies, and prodrugs [18,19,20,21,22]. However, the success of these compounds is compromised by the limitations of abnormal vasculature, heterogenic basement membranes, and poor blood supply. These are all inherited by TME, and are the causes of therapeutic failure [23,24,25]. Nanomedicine represents an important strategy to improve the delivery of therapeutic agents such as drugs, peptides, antibodies, proteins, genes, and immunotherapeutic agents in a selective and controlled manner for efficient accumulation and stimuli responsiveness [26,27,28,29,30,31,32,33,34]. Although great progress has been achieved in this field, the clinical translation of nanomedicines is still limited. In this review, we aim to present a discussion on the field of responsive nanomedicines. This will emphasize the application of biomaterials including natural polymers such as polysaccharides, biodegradable polymers, and metal oxides, in targeting the TME of solid tumors. Biomaterials represent a field of distinct research interest, due to their unique inherent properties. Their structure allows for effective functionalization for the co-delivery of multiple compounds and for effective responsiveness to an internal (chemical and/or biological) or external physical stimulus (magnetic field, light, radiation, ultrasound). Biomaterials have proved to be great supporters of theranostic applications in cancer treatment [35,36]. Overall, in this review we will discuss the role of biomaterial-based nanomedicines in targeting the TME. This includes heterogenic vasculature, tumor stroma ECM, CAFs, tumor hypoxia, and acidosis. We will examine the most recent advances in therapeutic nanomedicine for solid tumors; these have the potential to improve clinical outcomes. Finally, we will summarize the challenges and future outlook for the application of nanomedicines in tumor immunotherapy and combinational therapy to overcome limitations and improve the therapeutic outcome.

## 2. Solid Tumor Nanomedicine: Distribution in Tumor Microenvironment

Intriguingly, solid tumors are pathological organ-like tissues with heterogenic TME and increased metabolic status, which promote and support processes mimicking normal tissues, as angiogenesis [37,38,39]. Due to the elevated dysregulation of angiogenetic factors, abnormal and destabilized blood and lymphatic vessels are developed. These have major variations in diameter, density, shape (spiral-like) and overall distribution within TME. Additionally, a simultaneous discontinuation of endothelium with leaking cell gaps, irregularly thick or thin basement membranes and disruption of blood flow cause excessive spatial stress and increased interstitial fluid pressure (IFP) [40,41,42]. These features promote the transport of nutrients, oxygen, and blood away from the central region of solid tumors. This stimulates ATP regulation, hypoxia, and acidosis. The same features prohibit the transfer of therapeutic drugs, and this results in an inferior targeting effect and heterogenic tumor distribution of drugs [43]. Nanomedicines improve targeting effectiveness and selectivity [44,45,46]. This is especially the case when they are supported by an enhanced permeability and retention (EPR) effect, which promotes extravasation and effectual intratumor localization [47] (Figure 2). Despite progress being made, moderate clinical success has been achieved so far. This is because nanomedicine applications have encountered severe obstacles related to avascular tumor sites due to the TME’s characteristics restricting nanomedicines’ access only to highly vascular regions with increased perfusion [47,48]. These limitations in the therapeutic efficacy of nanomedicines have been tackled recently by exploiting the complex mechanisms and associated properties of TME. This improves intratumoral localization. Furthermore, specially designed nanomedicines exhibit stimuli-responsiveness in TME features, such as hypoxia and acidity, by combining ligand-mediated active targeting of selective receptors with growth factors, inhibitors, enzymes, and peptides. Combining the benefits of external stimuli-responsiveness has, beneficially, increased the targeting efficiency of nanomedicines and amplified the therapeutic activity [49,50,51,52].

Nanomedicines that target solid tumors traverse a highly variable environment. This emphasizes the importance of adaptability and spatiotemporal pharmacological activity. The current treatment strategies that aim to overcome MDR and the multi-factorial TME rely on a combination of chemo-, radio-, and immunotherapies (Figure 3). Approved nanomedicines are co-administered for improved synergistic therapeutic effect. In this way, biomaterials have greatly assisted in orchestrating the pharmacokinetics and targeting potential of nanomedicines. This is due to their inherent properties of biodegradability and ease of surface modification which offer effective sites for receptor-mediated ligand targeting (peptides, antibodies, nucleic acids, small organic molecules). Biomaterials also make possible theranostic applications of nanomedicines for real-time therapy and monitoring of tumor tissues. Moreover, biomaterials can be appropriately functionalized to obtain stimuli-responsiveness and manipulate TME hypoxia and spatial pH distribution (acidic extracellularly vs. basic intracellularly). In addition, the effectual combination of biomaterials with acquired responsiveness on varied external stimuli (hyperthermia, photodynamic, sonodynamic) has highly improved tumor-specific accumulation. This modulates cellular apoptotic cascades and promotes cellular death by associated gene regulation [53,54,55]. In this respect, biomaterials are important for TME-responsive applications, and they demonstrate promising results in combinational therapies and immunotherapies. Some of these are FDA-approved or under clinical trials for specific tumor types (Table 1) [56,57,58,59]. The clinical trials and applications of nanomedicines have mostly relied on improving drugs’ activity, as well as reducing side effects, and on enhancing biodistribution of drugs and agents within solid tumor tissues. For this, strategies have been developed to overcome physical, biological, and chemical obstacles in the TME, and achieve efficient specificity on molecular targets (such as cellular receptors, CAFs, CSC) and physiological factors (such as ECM, angiogenesis, IFP) [50,55,60]. The innate biological properties of biomaterials against opsonization have offered great support, because they provide longer and efficient systemic circulation time of the nanomedicines [61]. Natural polymers and synthetic biodegradable polymers which have been extensively reviewed and discussed for their application in the pharmaceutical and biomedical fields include: (i) polysaccharides such as alginic acid (alginate), dextran, agarose, hyaluronic acid, carrageenan, chitosan, and cyclodextrin: (ii) protein-based polymers such as gelatin, albumin, soy, and collagen; and (iii) synthetic polymers such as polyesters, polyamides, polyanhydrides, phosphorous based, and polyurethanes [62,63,64,65,66,67]. In solid tumor therapeutics, the benefits of novel theranostic and multifunctional nanomedicines are a niche research area that aims to overcome the limitations of TME and design novel therapies [67] (Figure 1). In the following, we discuss novel therapeutic and targeting concepts of biomaterial-based nanomedicines which exploit TME specificity and responsiveness.

**Table 1 pharmaceutics-16-00179-t001:** Therapeutic nanomedicines approved by the U.S. Food and Drug Administration (FDA) and the European Medicines Agency (EMA).

Carrier Type	Product Name	Therapeutic Agent	Cancer Type	Stage	Ref.
Liposomes	Zolsketil^®^	Doxorubicin	Metastatic breast cancer, advanced ovarian cancer, multiple myeloma, AIDS-related Kaposi’s sarcoma (https://www.ema.europa.eu/en/medicines/human/EPAR/zolsketil-pegylated-liposomal (accessed on 15 January 2024)	Approved (EMA, 2022)	[56,57]
	Vyxeos^®^	Cytarabine: daunorubicin	Newly diagnosed therapy-related acute myeloid leukemia, acute myeloid leukemia with myelodysplasia related changes (https://www.ema.europa.eu/en/medicines/human/EPAR/vyxeos-liposomal-previously-known-vyxeos, accessed on 15 January 2024)	Approved (EMA, 2018) (FDA, 2017)	[56,57]
	Onivyde^®^/CPX-351	Irinotecan	Pancreatic cancer (https://www.ema.europa.eu/en/medicines/human/EPAR/onivyde-pegylated-liposomal-previously-known-onivyde, accessed on 15 January 2024)	Approved (EMA, 2016) (FDA, 2015)	[56,57]
	Mepact^®^	Mifamurtide	Osteosarcoma (https://www.ema.europa.eu/en/medicines/human/EPAR/mepact, accessed on 15 January 2024)	Approved (EMA, 2009)	[56,57]
	Ameluz^®^	5-aminolevulinic acid	Superficial and/or nodular basal cell carcinoma (https://www.ema.europa.eu/en/medicines/human/EPAR/ameluz, accessed on 15 January 2024)	Approved (EMA, 2011)	[56,57]
	DaunoXome^®^	Daunorubicin	Kaposi’s sarcoma	Approved (FDA 1996) Discontinued (FDA, 2021)	[56,57]
Iron Oxide nanoparticles	NanoTherm^®^	Fe_2_O_3_	Glioblastoma, prostate, and pancreatic cancer (https://www.eib.org/en/stories/new-cancer-treatments, accessed on 15 January 2024)	Approved (EMA, 2013)	[57]
Albumin nanoparticles	Abraxane^®^	Paclitaxel	Metastatic breast cancer, locally advanced or metastatic non-small cell lung cancer, metastatic adenocarcinoma of the pancreas (https://www.ema.europa.eu/en/medicines/human/EPAR/abraxane, accessed on 15 January 2024)	Approved (EMA 2008) (FDA 2005)	[57,58]
	Pazenir^®^	Paclitaxel	Metastatic breast cancer, metastatic adenocarcinoma of the pancreas, non-small cell lung cancer (https://www.ema.europa.eu/en/medicines/human/EPAR/pazenir, accessed on 15 January 2024)	Approved (EMA 2019)	[58]
Vaccines	Adstiladrin^®^	Adenoviral vector-based gene therapy	Bacillus Calmette–Guérin unresponsive non-muscle invasive bladder cancer with carcinoma in situ with or without papillary tumors (https://www.fda.gov/drugs/resources-information-approved-drugs/fda-disco-burst-edition-fda-approval-adstiladrin-nadofaragene-firadenovec-vncg-patients-high-risk, accessed on 15 January 2024)	Approved (FDA 2022)	[59]
	Provenge^®^	Autologous peripheral-blood mononuclear cells	Metastatic castration-resistant prostate cancer (mCRPC) (https://www.drugs.com/history/provenge.html, accessed on 15 January 2024)	Approved (EMA 2013) (FDA 2010) Discontinued (EMA, 2015)	[59]

## 3. Nanomedicines for Targeting TME: Application of Natural and Synthetic Biomaterials

### 3.1. The Heterogenic Vasculature

Angiogenetic mechanism is divided into two phases: the avascular, wherein tumor progression is suppressed due to controlled homeostasis of pro- and anti-angiogenetic factors; and the vascular, wherein tumor development is promoted by a switched homeostasis favoring a pro-angiogenetic environment. For solid tumors, in order to progress and develop, a de facto ultimate need is presented for blood, oxygen, and nutrient supply. This is supported by the continuously evolving tumor vasculature (Figure 4) [68,69,70]. Thus, regulating angiogenesis is a key step in tackling TME abnormal vasculature that is being targeted by nanomedicines through angiogenetic inhibitors. The intention is to promote tumor suppression mechanisms by limiting the unrestricted vascular development [71]. The main target of angiogenetic therapeutics (Table 2) is the inhibition of growth factors of the pro-angiogenetic domain that present elevated affinity with surface receptors of ECs. These include: (i) soluble factors such as vascular endothelial growth factor (VEGF family factors comprising A to F members), platelet-derived growth factor (PDGF), beta and alpha transforming growth factors (TGF-α, -β), angiopoietins (Ang); and (ii) insoluble membrane-bound proteins such as ephrins, integrins, cadherins, matrix metalloproteinases (MMPs), and hypoxia-inducible factor-1 (HIF-1) (Figure 5) [72]. 

#### 3.1.1. VEGF Therapeutic Targeting

VEGF is the most widely researched target of anti-angiogenetic therapeutics. This is because the VEGF/VEGFR2 signaling cascade is a crucial regulator promoting angiogenesis, vascular permeability, proliferation, and migration [68,69,70,71,72,73]. Among angiogenesis inhibitors, heparin has been studied in cancer nanomedicine. This is due to its strong anticoagulation effects that indicate enhanced binding affinity with VEGFR2. Heparin is an FDA-approved anticoagulant drug. However, its implementation against malignant tumors present limitations related mainly to severe side effects such as bleeding and thrombocytopenia. The problem in heparins’ application has been tackled by utilization of low-molecular-weight heparins (LMWHs) or heparin analogues [74]. Other angiogenesis inhibitors in the category of biomaterials include the sulfated polysaccharides such as hyaluronic acid (HA) and chitosan (CS). These have expressed selective binding affinity with VEGFR receptors and structural similarities with heparin. Lim et al. [75] studied the inhibition effect of sulfated HA (s-HA) against VEGF_165_ factor, which is present in two isoforms: the VEGF_165a_ angiogenetic and the VEGF_165b_ anti-angiogenetic. Both of these express the same receptor-binding domain (RBD) but a different heparin-binding domain (HBD). The density of HA sulfation was varied from one hydroxyl group per repeating unit (this resulted in strong affinity of VEGF_165a_) to more than one (up to four; this promoted a strong binding affinity to both VEGF_165_ isoforms. The s-HA was reported to provide comparable properties to the commercially used anti-VEGF antibody Avastatin for the non-selective binding of VEGF_165_ [75]. In another study, Li et al. [76] demonstrated that sulfated CS (s-CS) could act as an angiogenetic inhibitor by blocking the VEGF/VEGFR2 signaling pathway. The degree of CS sulfation at C2–NH_2_, C3–OH, and C6–OH sites per repeating unit could be related to the mechanism of angiogenetic inhibition. In contrast to heparin, s-CS presented a stronger inhibitory effect on proliferation, migration, and tube formation of HUVEC cells. This promoted tumor size inhibition by nearly 42.12% and suppressed neovascularization by almost 63.8% [76].

Biomaterials were used in combination with angiogenetic inhibitors over theVEGF/VEGFR signaling pathway. Among biomaterials, great advantages are demonstrated by CS. In addition to its antitumor effect, this includes inhibition of proliferation, induction of apoptosis, and improvement of immune functions. In this respect, Salva et al. [77] studied the effect of CS nanoplexes with small interfering RNA (siRNA) targeting VEGF expression on the suppression of tumor growth. CS nanoplexes with different siRNAs, such as siVEGF-A, siVEGFR-1, and siVEGFR-2 were evaluated for their silencing effect upon intratumoral injection in Sprague Dawley rats bearing breast tumor. The CS-siRNA nanoplexes resulted in remarkably reduced tumor volume and mRNA levels of VEGF in tumor tissues. In another study, Jiang et al. [78] investigated the antitumor and anti-angiogenetic effect of carboxymethyl chitosan (CMCS). CMCS was found able to constrain the 2D and 3D migration of HUVEC cells in vitro and promote tumor growth inhibition and tumor cell necrosis in vivo in hepatocarcinoma 22 (H22) tumor-bearing mice. Moreover, CMCS regulated the expression of VEGF, MMP-1 and CD34 levels in the serum and H22 tumor tissues, and improved the thymus and spleen index, TNF-α, and IF-γ levels. Overall, CMCS significantly promoted inhibition of tumor angiogenesis and stimulated immune functions.

#### 3.1.2. Targeting Molecular Markers for Vasculature Regulation

Another promising therapeutic target that can effectively promote suppression of tumor angiogenesis is pericytes; these are a category of mural cells (MCs). Pericytes play a key role in important intracellular interactions of ECs including proliferation, migration, and stabilization of angiogenesis and vascular basement membrane. Thus, pericytes act as angiogenetic regulators while ECs promote proliferation of pericytes by acting as angiogenetic stimulators [79]. In normal angiogenesis, sprouting ECs, by secreting PDGF, stimulate the proliferation of MCs such as pericytes, which are positive on the PDGF β receptor (PDGFR-β). This further promotes the release of VEGF-A cytokine and Ang-1 protein, and this stabilizes and maintains vascular development. Cytokine secretion by the VEGF family can effectively mediate multiple signaling pathways such as the paracrine signaling cascade, to orchestrate direct and indirect interactions between ECs and pericytes. This results in the regulation of cytokines, enzymes, and receptors such as TGF-β, MMP, and Tyrosine Kinase-2 (Tie2) receptors [80,81,82]. However, in tumor tissues the interactions between ECs and pericytes are defective, and this results in abnormalities in the structure, regulation and density of pericytes. This contributes to increased heterogeneity on vasculature and triggers a hypoxic TME. Due to their dynamic characteristics and multiple molecular markers expression, pericytes in TME have a great potency to differentiate cancerous stromal fibroblasts that contribute to invasion and metastasis [80,81,82].

The therapeutic potency of tumor cells’ molecular markers has led to intense research on targeting of the heterogenic vasculature of solid tumors [83,84]. Bhattacharya et al. [85] reported the development of HA-conjugated Pluronic^®^ P123/F127 copolymer nanoparticles (HA-TQ-NPS) for the selective delivery of thymoquinone (TQ) to TNBC cells. The synergistic effect of HA and TQ efficiently hindered cell migration by modulating the expression of miR-362/Rac1/RhoA and miR-361/VEGF-A pathways. The latter was involved in the suppression of tumor angiogenesis. The evaluation of the HA-TQ-NPS nanoparticles in MDA-MB-231 xenograft chick embryos and 4T1 tumor-bearing BALB/c mice demonstrated their ability to inhibit angiogenetic and metastatic activities. The combinational silencing of PDGF and their receptors was followed by an investigation by Salva et al. [86] into the ability to suppress multiple members of the same pathway. In this study, CS nanoplexes were used to deliver dual and single siRNA targeting PDGF-D and PDGFR-β expressions. The intratumoral administration of the nanoplexes in breast tumor-bearing mice xenografts resulted in the inhibition of tumor growth and angiogenesis. This prevented invasion and further downregulating of PDGF-D/PDGFR-β mRNA and proteins’ level expression. Another type of nanoparticle that is indicative of bioactive ceramics that have been proposed for their anti-angiogenetic properties are hydroxyapatite nanoparticles (HANPs). In a recent study, Shi et al. [87] demonstrated the biological interactions of HANPs with endothelial HUVEC cells to provide insight into the suppressive effect of the nanoparticles on angiogenesis. This is achieved through regulating ECs function via the PI3K/Akt/eNOS signaling pathway. Specifically, HANPs suppressed the phosphorylation of eNOS and p-eNOS, and this resulted in the downregulation of p-Akt. In another study by Zhao et al. [88], the development of amine-functionalized HANPs was reported for the combinational delivery of p53 plasmid and candesartan (CD). This is an inhibitor with strong affinity to angiotensin II type 1 receptor (AT_1_R). The combined anti-angiogenetic and antitumor efficacy of the nanoparticles was reported by the downregulation of VEGF protein secretion and lower functional microvessel density. Tyrosine kinase inhibition activity has also been tested for antitumor and anti-angiogenetic effectiveness by Garizo et al. [89]. A p28 (28 amino acids, 28 kDa) cell-penetrating peptide (CPP) was surface conjugated on poly (lactic-co-glycolic acid) (PLGA) nanoparticles loaded with gefitinib (GEF). GEF is a tyrosine kinase inhibitor with poor bioavailability and therapeutic activity due to its weak solubility in gastric fluids. The PLGA nanoparticles were evaluated in female CBA/N mice bearing A549 lung adenocarcinoma tumors which express the EGFR. The nanoparticles promoted the inhibition of both primary tumor growth and metastatic burden by combining selective accumulation with the tyrosine kinase-inhibiting activity of gefitinib.

#### 3.1.3. Targeting Formulation Based on FDA-Approved Drugs

The extensive research on anti-angiogenetic therapies has led to the development of various agents which have already been approved by the FDA and the EMA or are under clinical investigation. For example, bevacizumab (VEGF-A antibody), ramucirumab (VEGFR2 antibody), aflibercept (VEGF-Trap), lenvatinib (Tie2 inhibitor), sorafenib (Tie2 inhibitor), and axitinib (Tie2 inhibitor) [90,91,92]. These medicines have been applied mainly as secondary or adjuvant therapies to prohibit tumor vascularization and suppress tumor growth. Within the research field of anti-vascular therapies, the key function of the VEGF/VEGFR signaling pathway in tumor angiogenesis and metastasis is well studied. Goel et al. [93] reported the efficacy of mesoporous silica nanoparticles (MSNs) for targeting the delivery of sunitinib, a tyrosine kinase receptor inhibitor that is able to multi-target almost every VEGFR. Sunitinib is a FDA-approved anti-VEGFR drug for renal cell carcinoma (RCC) and imatinib-resistant gastrointestinal stromal tumor (GIST). In a study by Goel et al., pegylated sunitinib-MSNs were modified with NOTA chelator agent, and further linked with VEGF_121_ and radioisotope (^64^Cu, t_1/2_ = 12.7 h) labeling. The sunitinib-MSNs were evaluated for their theranostic application against U87MG human glioblastoma-bearing athymic nude mice. The overall in vitro, in vivo, and ex vivo analyses confirmed (i) the increased VEGFR targeting specificity in the vasculature regions (through CD31 staining), (ii) the effective sunitinib accumulation to tumor sites for efficient inhibition, and (iii) the ability of angiogenesis imaging. Bevacizumab is a water soluble recombinant monoclonal anti-VEGF antibody that is effective in inhibiting tumor angiogenesis. It is administered in combination with chemotherapy. However, bevacizumab is associated with serious side effects such as cardiovascular disorders, hypertension, thromboembolic, and central nervous system hemorrhage. Thus, the application of bevacizumab in nanomedicine has been highly researched to investigate how to alleviate the side effects. Abdi et al. [94] reported the synthesis of lipid-coated chitosan nanoparticles for the local administration of bevacizumab (BEV). This resulted in enhanced antitumor activity by suppressing proliferation and ECs angiogenesis. In another research by Balao et al. [95], PLGA-PEG nanoparticles delivering bevacizumab (BEV) were surface functionalized with a targeting antibody fragment (Fab, AbD15179) with increased binding affinity over the CD44v6 cellular receptor. The BEV-loaded PLGA-PEG nanoparticles were evaluated against colorectal cancer (CRC) cells that overexpress the CD44v6 by nearly 50%, and were found to exhibit higher internalization into CD44v6+ ECs than the bare nanoparticles; there were also elevated intracellular levels of bevacizumab and VEGF. The CD44v6 transmembrane protein is a CD44 receptor isoform containing exon v6. This is important for solid tumors metastasis and invasion due to its function for c-Met, VEGFR-2, and angiogenesis. The efficacy of BEV has also been studied by Luis de Redin et al. [96], using PEG-coated human serum albumin (HSA) nanoparticles encapsulating BEV. The nanoparticles were evaluated on HT-29 colorectal cancer xenograft models in athymic nude mice and were found to exhibit increased binding affinity on ECs receptors. This is due to the innate properties of HSA, which resulted in elevated intratumoral localization of BEV. Moreover, encapsulated BEV resulted in decreased glycolysis, reduced BEV blood levels, and reduced metabolic tumor volume and angiogenesis (evaluated by CD31 staining) compared to free BEV administration.

Sorafenib (SF, Nexavar^®^ antineoplastic agent Bayer AG, Leverkusen, Germany) is another FDA-approved agent against liver cancer, thyroid cancer, and hepatocellular carcinoma (HCC). Sorafenib is a kinase inhibitor with the activity of inhibiting cancer cell proliferation and angiogenesis by targeting and blocking the action of the multiple Raf family kinase proteins (B-Raf and C-Raf), the VEGFR-2 and the PDGFR, and their associated signaling cascades of the ERK pathway. However, the hydrophobic nature and poor solubility of sorafenib in deionized water (25 mg/mL) limits the daily administered dose. Thus, sorafenib tosylate (Nexavar^®^; the organosulfonate salt of sorafenib) is also an FDA-approved synthetic compound targeting growth signaling and angiogenesis. It is administered against renal, hepatocellular, and differentiated thyroid cancers in daily doses up to 400 mg (2 pills of 200 mg tablet per diem). Ruman et al. [97] reported the development of folate-coated CS nanoparticles for the delivery of SF. The SF-loaded nanoparticles exhibited high antitumor activity against hepatocellular carcinoma cells and colorectal adenocarcinoma cells. Sorafenib has been extensively investigated in nanomedicine delivery systems. These are designed to overcome the limitation of poor solubility, non-specific targeting and drug resistance of free SF. Such SF nanomedicine systems have been evaluated in co-delivery with biomaterials, chemotherapeutic drugs, imaging agents, and active agents for receptor-mediated binding. They have also expressed great potential in application against various solid tumors, as has been recently reviewed by Wang et al. [98]. 

#### 3.1.4. Responsive Targeting and Combinational Therapies

For the efficient targeting of heterogenic vasculature, unprecedented growth has been demonstrated in functionalized biomaterials for multi-responsive nanomedicine applications. Du et al. [99] studied the effect of gambogenic acid (GNA). This is delivered by charge-reversible polymeric nanoparticles (CRNP) of (PEG_45_-PCL_40_-PAEA_33_-SA) for increased vascular permeability and improved drug accumulation in the tumor. GNA inhibits VEGF and shows an increased antitumor effect due to its pro-apoptotic activity; and vascular disruption due to the downregulation of angiogenesis pathways. Thus, the CRNP-GNA particles resulted in increased vascular permeability and retention indexes (VPRI) by near 60 times and decreased tumor microvessel density by nearly 7%, compared to their charge-irreversible analogue. The CRNP-GNA particles achieved effective intratumor accumulation in tumor-bearing C57BL/6 mice models. This is due to effective vascular disruption and suppressed angiogenesis, with low to no presence of vascular tubes inside the tumor. The combinational therapies and mechanisms applied in regulating angiogenesis and metastasis of non-small cell lung cancer solid tumors have been studied by Zhang et al. The study investigated the combinational effect of gefitinib and bevacizumab delivered by MnO_2_-containing liposomes [100]. Another example has been provided by Punuch et al. [101], who investigated the synergistic effect of angiogenesis inhibitors (sorafenib or sunitinib) with AFP-specific siRNA, incorporated into polymeric PLGA nanoparticles. The combinational effect resulted in the effective inhibition of cell proliferation of hepatocellular carcinoma. The strategy of carrier multifunctionality was followed by a study by Cong et al. [102] into the efficient delivery of anlotinib. Anlotinib is a multi-targeting inhibitor of tyrosine kinase receptors including VEGFR, FGFR, PDGFR, and c-Kit, and it has increased inhibitory affinity on tumor angiogenesis. Cong et al. [102] applied pH-responsive PLGA-PEOz polymers mixed with PLGA-PEG co-polymers. The anti-angiogenic mechanism of anlotinib was combined with the pH-responsive PLGA-PEOz/PLGA-PEG nanoparticles for systemic administration in BALB/c nude mice bearing A549 and 4T1 tumor xenografts. The effective anlotinib targeting successfully inhibited tumor growth and metastasis by suppressing lymphangiogenesis through VEGFR-3 signaling cascade. Cheng et al. [103] combined the effects of siRNA silencing of VEGF expression with doxorubicin in a nanomedicine system of polycation liposomes encapsulating calcium phosphate nanoparticles. This significantly suppressed tumor growth and angiogenesis in MCF-7 tumor-bearing BALB/c nude mice. Multi-targeting effects were also reported by Barui et al. [104] in liposomes loaded with curcumin and doxorubicin, and surface functionalized with a pegylated RGDK-lipopeptide for ECs targeting. The combinational system co-delivered a chemotherapeutic agent with curcumin, which is well known for its inhibitory effect on the activation of NFκB transcription factor linked to chemoresistance, and on ATP-binding cassette drug transporter. Their synergistic action with the targeting lipopeptide suppressed tumor growth, invasion, and metastasis-related genes at mRNA and protein levels.

In combinational and multi-responsive nanomedicine applications, metal nanoparticles have been of great research interest due to their angiogenetic properties. Most of the metal nanoparticles including gold, silver, zinc oxide, and copper, have been found to participate in the angiogenesis process by regulating the expression of ROS and reactive nitrogen species (RNS) at a cellular level. The inhibition of vascular development is based on the ability of the metal nanoparticles to regulate the balance of secreted pro- and anti-angiogenetic factors in order to promote apoptotic signaling mechanisms [105,106]. The selective inhibitory effect and modulation of angiogenesis were studied by Roma-Rodrigues et al. [107] with peptide conjugated gold nanoparticles in chick embryos. Moreover, Bartczak et al. [108] reported the effect of gold nanorods functionalized with peptides selectively binding to neuropilin-1 surface receptors (NRP-1), in combination with laser irradiation. In the combinational system, gold nanorods absorbed near-infrared laser irradiation and transformed it to localized heat and the peptide acted as an angiogenetic inhibitor. By modulating the metal nanorods dose and NIR irradiation, effective inhibition of in vitro angiogenesis was promoted. 

**Table 2 pharmaceutics-16-00179-t002:** Nanomedicines based on biomaterials targeting heterogenic vasculature.

Targeting Effects	Carrier Type	Therapeutic Agent	Characteristics	Ref.
VEGF	Hydroxyapatite (HA)	Sulfated s-HA	Non-selective binding of VEGF_165a_	[75]
	Chitosan (CS)	Sulfated s-CS	Inhibition of VEGF/VEGFR2 signaling pathway	[76]
	CS/siRNA nanoplexes	siRNA	Silencing effect of siVEGF-A, siVEGFR-1, siVEGFR-2, and NRP-1 inhibiting proliferation with improved immune functions	[77]
	Carboxymethyl chitosan (CMCS)	CMCS	Regulate dexpression of VEGF levels, MMP-1, and CD34, and promoted inhibition of angiogenesis	[78]
Endothelial Cell Regulation	HA-P123/F127 Polymeric nanoparticles	Thymoquinone	Modulating expression of miR-362/Rac1/RhoA and miR-361/VEGF-A pathways for inhibiting angiogenesis	[85]
	CS nanoplexes	siRNA	Targeting PDGF-D and PDGFR-β expressions	[86]
	Hydroxyapatite nanoparticles (HANP)	HANP	Regulating ECs function by the PI3K/Akt/eNOS signaling pathway	[87]
	Hydroxyapatite nanoparticles	p53 plasmid and candesartan	Downregulation of VEGF protein secretion and functional microvessel density	[88]
	PLGA nanoparticles	P28 peptide and gefitinib	Inhibit tumor angiogenesis, primary tumor growth, and metastasis	[89]
FDA-Approved Drugs	Mesoporous silica nanoparticles (PEG-MSNs)	Sunitinib (anti-VEGFR)	Increased VEGFR targeting specificity, efficient inhibition of angiogenesis	[93]
	Lipid-chitosan nanoparticles	Bevacizumab (VEGF-A antibody)	Suppressing proliferation and endothelial cells angiogenesis	[94]
	PLGA-PEG nanoparticles	Bevacizumab	Higher internalization and bevacizumab delivery into CD44v6+ ECs	[95]
	Human serum albumin nanoparticles	Bevacizumab	Decreased glycolysis and metabolic tumor volume, inhibition of tumor growth	[96]
	Chitosan nanoparticles	Sorafenib (Tie2 inhibitor)	Superior antitumor activity	[97]
Combinational Therapies	PEG-PCL-PAEA-SA nanoparticles	Gambogenic acid/charge-reversible effect	Suppressed tumor angiogenesis, very little to no vascular tubes inside tumor models	[99]
	PLGA nanoparticles	Sorafenib/Sunitinib/siRNA	Synergistic effect inhibiting cell proliferation	[101]
	PLGA-PEG nanoparticles	Anlotinib/pH-sensitivity	Inhibited tumor growth and metastasis suppressing lymphangiogenesis	[102]
	Polycation liposomes	siRNA/calcium phosphate particles	Suppressed tumor growth and angiogenesis	[103]
	PEG-liposomes	Doxorubicin/curcumin	Suppressed tumor growth, invasion, and metastasis	[104]
	Au nanorods	NRP-1 peptide/PDT	Inhibition of angiogenesis	[108]

### 3.2. The Tumor Stroma Extracellular Matrix

Therapeutic approaches against solid tumors expand on targeting tumor stroma (Figure 6). This is a dynamic heterogenic matrix that commonly comprises cellular components such as cancer-associated fibroblasts (CAFs), mesenchymal stromal cells, innate and adaptive immune cells, macrophages, and non-cellular compartments such as extracellular matrix (ECM), tumor vasculature, and interstitial matrix [109,110]. Tumor stroma has been recognized as a critical part of TME because its abundant components can support the transformation of normal cells to tumorous cells, promoting tumorigenesis and progression [111]. In this process, ECM plays a pivotal role in the development and support of homeostasis of tumor tissue. ECM is mainly composed of proteins (such as collagen, fibrilin, and elastin), proteoglycans, and polysaccharides (hyaluronic acid or hyaluronan). The unrestrained cross-linking of hyaluronic acid with collagen type I results in a densely rigid network. This represents a critical obstacle for host immune systems’ function [112]. Tumor ECM has dynamic features and a pronounced effect on supporting TME vascularization [113]. It also (i) presents a physical barrier for drug perfusion; (ii) induces cell adhesion-mediated drug resistance owing to the elevated presence of ECM proteins; (iii) activates glycolysis and glutamine metabolism to provide an abundant energy supply for tumor cells; (iv) promotes aggressiveness and metastasis by the deposition of malignant cells to the tumor endothelium [112,113,114]. Various factors participate in ECM maintenance, including growth factors (VEGF, PDGF, Ang, stromal cell derived factor, SCDF) and proteins (MMP family). These factors are the target of ECM-specific nanomedicines (Table 3) [115].

#### 3.2.1. Hyaluronidase for ECM Targeting

The degradation of ECM components that are responsible for the increased adhesion of tumor cells has been targeted therapeutically by nanomedicines. The most characteristic example of this being the PEGylated recombinant human hyaluronidase enzyme, PEGPH20, which is designed to inhibit HA cross-linking. PEGPH20 has been tested as a monotherapy and in combination with chemotherapeutic drugs. This has produced diverse results; for example, in clinical trials there were cases of patients undergoing the therapeutic scheme of PEGPH20 experiencing deterioration of the therapeutic outcome and side effects such as muscle spasm and thromboembolism [115,116]. PEGPH20 is a multiple PEGylated recombinant HyAL5 (recombinant human hyaluronidase PH20, or rHUPH20) with a half-life of 1 to 2 days. It has been clinically evaluated in a phase III trial (HALO-109-301) with advanced pancreatic cancer patients in a combinational therapeutic scheme with gemcitabine and nab-paclitaxel [117]. HA is a strongly hydrophilic anionic, non-sulfated glycosaminoglycan (GAG) with physiological functions in cells’ activation and proliferation via interaction with surface receptors such as CD44 and RHAMM. There are five hyaluronidases (HyAL1-5); these are endogenous enzymes that are able to degrade HA into oligosaccharides and very low-molecular-weight HA. In this way, they facilitate drug penetration into the tumor. The degradation products of HA transmit inflammatory responses through toll-like receptor 2 and 4 (TRL2, TRL4) in macrophages and dendritic cells. This plays a pivotal role in innate immunity. Thus, hyaluronidase and other ECM-degrading enzymes have been included in nanomedicine and biomaterials research in order to overcome the biological barriers which limit their systemic administration and therapeutic efficacy. These limitations include short half-life (of a few minutes) in the bloodstream, inactivation and loss of enzyme function, and side effects such as breakdown of ECM in healthy tissues due to non-tumor specificity. Thus, Zhou et al. [118] studied the conjugation of rHUPH20 in doxorubicin-loaded PLGA-PEG nanoparticles functionalized with an extra PEG layer for protection of hyaluronidase. It was reported that the conjugated rHUPH20 was more effective than the free enzyme, and this significantly increased tumor penetration and accumulation of nanoparticles in 4T1 tumor-bearing syngeneic BALB/c mice. This effect was attributed to the extra PEG layer that reduced the exposure of rHUPH20 to the in vivo environment. The increased tumor accumulation resulted in an enhanced antitumor effect of the doxorubicin nanoparticles.

#### 3.2.2. Extracellular Matrix Degradation

Degradation of the extracellular matrix of TME has been considered as a means for enhancing the transportation of therapeutic agents and drugs. The enzymatic degradation of ECM was studied by Ikeda-Imafuku et al. [119]. They applied bromelain bioenzyme covalently conjugated through a hyaluronic acid (HA) linker with C4BP targeting peptide having increased binding affinity with collagen type IV of ECM. Bromelain is a bioenzyme with enhanced ECM proteolysis activity over a wide range of pH and ECM proteins, and is utilized to enhance the efficacy of chemotherapy. The administration of bromelain-HA-C4BP conjugates resulted in effective tumor accumulation in 4T1 tumor-bearing mice, and decreased collagen fibers’ density. This enhanced the biodistribution of doxorubicin liposomes (Doxosome). Polymeric PLGA-based nanoparticles were investigated by Amoozgar et al. [120] in a combinational study for promoting ECM degradation and enhancing antitumor activity of doxorubicin. The PLGA nanoparticles were coated with polydopamine acting as an adhesive layer, for further modification with PEG and proteins including lysozyme, DNase, collagenase I, and E-selectin antibody. Administration in 4T1 breast tumor-bearing BALB/c mice showed that the PLGA-collagenase I nanoparticles effectively promoted degradation and enhanced the intratumoral distribution of doxorubicin. This further enhanced antitumor immunity because doxorubicin was present in immunosuppressive M2 macrophages. Proliferation of lymphocytes was also observed. Bioenzymes such as hyaluronidases and collagenases catalytically promote the degradation of ECM substrate molecules. These have been widely investigated in nanomedicine, as reported in a recent review by Ding et al. [121]. ECM-targeting therapeutics are based on degradation processes to disrupt the cross-linking and stabilization of ECM proteins. In this mechanism, the inhibition of lysyl oxidase (LOX) activity has been highly studied because LOX acts as a catalyst to facilitate collagen cross-linking in particular. Among LOX inhibitors, simtuzumab (LOXL2 antibody), PAT-1251 (LOXL2 inhibitor), and PXS-5382 A (LOX inhibitor) have been evaluated as adjuvant therapies in various phases of clinical trials. This is because, as monotherapies, these have not improved the clinical outcome [122]. The effect of LOX inhibition on ECM morphology was investigated by Grossman et al. [123] by screening stages of fibrillary collagen assembly through targeting LOXL2. The therapeutic efficacy of disrupting LOXL2 was further evaluated in tumor-bearing CB-17 SCID mice, and this resulted in the maintenance of normal fibril orientation and thickness. This potentially affected tumor progression since LOXL2 levels are associated with collagen assemblies at the nanoscale, and fiber orientation at the microscale.

The combinational delivery of trastuzumab (Herceptin), a HER2-targeted monoclonal antibody, and collagenase were investigated by Pan et al. [124]. They used PLGA-PEG-PLGA polymeric thermosensitive hydrogels for peritumoral administration in HER2+ BT474 tumor-bearing BALB/c mice. The synergistic administration caused degradation of the collagen fibrils in the intratumoral region, and this further promoted the antibody’s effect. LOXL2-targeted mPEG-PLGA polymeric nanoparticles loaded with DDR1 inhibitor were studied by Wei et al. [125] to effectively remodel tumor stroma in vitro and in vivo in KPC/M-PSC orthotopic xenografts with desmoplastic stroma. The mPEG-PLGA-DDR1 polymeric nanoparticles were further encapsulated in peptide liposomes for the co-delivery of LOXL2 inhibitor. The polymer–lipid–peptide nanoparticles were applied as a first-line system targeting the extracellular LOXL2 and the intracellular DDR1 domain in tumor stroma cells. The system effectively suppressed collagen cross-linking and MMP1 secretion, promoting the remodeling of stromal arrangement. In a second administration step, Nab-paclitaxel was delivered across the remodeled stroma; this enabled enhanced penetration and accumulation in KPC/M-PSC tumors.

Chen et al. [126] studied the effect of MMP9 responsive magnetic nanoparticles for effective bioimaging and synergistic chemo-photothermal therapy. The ultrasmall superparamagnetic iron oxide (USPIO) nanoparticles loaded with doxorubicin were further encapsulated in self-assembling amphiphilic PEGylated polypeptides that expressed MMP9-sensitivity. The micelles were evaluated in BALB/c 4T1 tumor-bearing mice. The MMP9-sensitive polypeptide chains rapidly degraded in the tumor, and released USPIONs and doxorubicin. Moreover, USPIONs demonstrated enhanced ability as T2-T1 switching MRI contrast agents. In combination with laser irradiation, the nanoparticles exhibited good chemo-photothermal antitumor effect.

#### 3.2.3. Targeting Biomolecules for Extracellular Matrix

Intense research on biological mechanisms has promoted the synthesis of ECM biomimetic materials as a potential strategy for effective inhibition of tumor metastasis and invasion by offering a network that functions as a physical barrier for cell migration. This was reported by Guo et al. [127] for agarose hydrogel cell adhesive micropatterns. In this context, Hu et al. [128] developed an in situ artificial ECM (AECM) as a barrier for tumor cell migration, based on transformable laminin (LN)-mimetic peptide nanoparticles. The laminin biomimetic peptide was composed of peptide amino-acid sequences such as RGD and YIGSR, and was capable of self-assembling to form nanoparticles. Upon intravenous administration in lung-metastasis tumor-bearing mice, the laminin nanoparticles were transformed into nanofibers, which surrounded the tumor site and formed an AECM to effectively suppress lung metastasis. Laminin is a family of glycoproteins that are of critical importance for ECM. This is because it is a main constituent of the basal lamina membranes’ layer. Laminin with fibronectin and collagen represent adhesion proteins that are able to bind to integrins on the cellular surface through specific cell-binding domain epitopes. These epitopes are small amino-acid peptide sequences, and include RGD, RGDS, IKVAV, and YIGSR [129].

Another strategy for evaluating biomolecules for ECM targeting is the blockade between ECM and cellular interactions by inhibition of signaling molecules such as the proteins that regulate ECM deposition; this includes the connective tissue growth factor (CTGF). Such therapeutic strategies are actively being studied in clinical trials; and TGF-β, and integrin targeting molecules, FAK-inhibitors, and Pamrevlumab anti-CTGF antibody therapy are currently being evaluated in phase III clinical trials [130]. Ding et al. [131] examined the effect of siRNA on suppressing CTGF expression in MDA-MB-231 in vitro and in vivo tumor-bearing mice. Hyaluronic acid was coated on mesoporous silica nanoparticles, (MSNs) conjugated with a PEGA-pVEC cell penetration peptide and further delivering siRNA and doxorubicin in order to provide RES protection and targeting ability on CD44 receptors, which are overexpressed in breast cancer cells. Upon administration, the targeting peptide enabled effective accumulation of the nanoparticles at the tumor site and HA provided increased cellular internalization through CD44-mediated endocytosis. Hyaluronidase in the lysosomes activated the degradation of HA coating, and this enabled responsive doxorubicin release and drug-induced apoptosis. Moreover, the release of siRNA effectually suppressed protein expression levels, and this plays a key role in multidrug resistance, such as Bcl-xL, cIAP1, and CTGF; this further promoted tumor cells’ susceptibility to apoptosis. 

**Table 3 pharmaceutics-16-00179-t003:** Nanomedicines based on biomaterials targeting stroma extracellular matrix.

Targeting Effects	Carrier Type	Therapeutic Agent	Characteristics	Ref.
Hyaluronidase	PEGPH20	PEGPH20/gemcitabine/nab-paclitaxel	Phase III trial (HALO-109-301)	[117]
	PLGA-PEG nanoparticles	rHUPH20/doxorubicin	Effective tumor accumulation enhanced antitumor effect	[118]
ECM Degradation	Doxorubicin liposome (Doxosome)	Bromelain/Hyaluronic acid linked collagen type IV-binding peptide	Decayed the density of collagen fibers and advanced the tumor distribution	[119]
	PLGA-polydopamine-PEG nanoparticles	Collagenase I/Doxorubicin	Degradation, enhanced the intratumoral distribution, and enhanced antitumor immunity	[120]
	LOXL2 antibody	LOXL2 antibody	Control of collagen assembly in ECM, potentially control tumor progression	[123]
	PLGA-PEG-PLGA thermosensitive hydrogel	Trastuzumab (Herceptin)/collagenase	Degradation of intratumoral collagen promoting the antibody effect	[124]
	mPEG-PLGA nanoparticles	LOL2 and DDR1 inhibitors/Nab-paclitaxel	Enhanced penetration and accumulation in tumor	[125]
	Ultrasmall superparamagnetic iron oxide (USPIO) nanoparticles	MMP9-sensitive peptide/Doxorubicin	Effective bioimaging and synergistic chemo-photothermal antitumor effect	[126]
ECM Biomolecules	Peptide nanoparticles	Laminin (LN) mimic peptide	Increased distribution in the tumor site and simultaneous transformation into nanofibers surrounding the tumor site	[128]
	Hyaluronic acid mesoporous silica nanoparticles	siRNA suppressing CTGF expression/Doxorubicin	Inhibition of multidrug resistance and increased susceptibility of tumor cells to drug-induced apoptosis	[131]

### 3.3. The Tumor Stroma Cancer Associated Fibroblasts

Tumor stroma is a dynamic environment that orchestrates tumor progression, angiogenesis, invasion, and metastasis through various components, including stromal cells such as cancer-associated fibroblasts (CAFs), mesenchymal stem cells (MSCs), ECs, stellate cells, and adipocytes (Figure 7). CAFs are activated fibroblasts that, in contrast to normal fibroblasts, have established a prevailing role in (i) the increased rate of proliferation and migration, (ii) the ability to regulate tumorigenesis, (iii) the development of inter- and intracellular interactions with tumor cells, (iv) metabolic reprogramming of tumor cells during tumor initiation, (v) energy supply through oxidative phosphorylation for sustaining the elevated proliferation rate, and (vi) autophagy and oxidative stress pathways [132]. CAFs targeting is a strategically crucial therapeutic field of TME-intervening nanomedicines (Table 4). This is because CAFs are related to drug resistance by secreting proteins, cytokines, and extracellular vehicles that protect malignant tumor cells and make TME hostile to antitumor agents and host immune attack [133,134,135]. The essential role of CAFs in tumorigenesis, multidrug resistance (MDR), and metastasis has been addressed in this review by responsive biomaterials applied on the targeting of cellular surface markers for CAFs depletion, and on the normalization of activated CAFs for phenotype reprogramming. Among surface biomarkers, the most distinguishing and well-researched are fibroblast activation protein-α (FAP-α), which represents a poor prognostic marker [136]; and α-smooth muscle actin (α-SMA), which has been identified as a novel biomarker [137]. FAP is a transmembrane glycoprotein (antigen) that is present in almost 90% of stromal fibroblasts, and its effective targeting and suppression induced CD8^+^ T cell-mediated damage of CAFs. This results in inhibition of tumor growth [135,136,138].

#### 3.3.1. Targeting Nanomedicine for CAFs Depletion

The targeting of surface biomarkers has been highly involved in CAFs targeting for promoting their depletion. Specifically, FAPs biomarker has been studied as a targeting domain for tumor imaging and diagnosis by Ruger et al. [139]. They developed fluorescence liposomes (anti-FAP-IL liposomes) conjugated with single-chain specific-fragments (Fv) directed against FAP (scFv’FAP). These presented increased binding affinity to FAP-overexpressing tumor cells in tumor-bearing athymic nude mice (HT1080-wt, HT1080-hFAP, MDA-MB435S tumors). This resulted in effective NIR fluorescence imaging, where the drug had selectively accumulated at tumor sites via FAP. CAFs specific targeting was also approached by cleavable amphiphilic peptides (CAP) that can specifically and efficiently respond to FAP-α cell surface biomarker. Ji et al. [140] studied the amphiphilic nature of CAP monomers containing a TGPA peptide sequence being cleaved by FAP-α. The CAP monomers self-assembled to nanofibers that, in the presence of doxorubicin, further induced their assembly to nanoparticles. The CAP nanoparticles (CAP-NP) disassembled upon FAP-α cleavage, promoting the rapid local release of doxorubicin in prostate tumors by disturbing the stromal barrier and increasing drug intratumoral accumulation. FAP-α targeting was also studied by Yu et al. [141] in pancreatic Pan 02 subcutaneous and orthotopic tumor-bearing C57BL/6 mice, for combined chemo- and photothermal therapy. The multi-targeting biomaterial system was evaluated against pancreatic tumors, which are characterized by a dense and strong tumor stroma generating a physical barrier to drug delivery. Thus, CAP peptide self-assembling thermosensitive liposomes (CAP-TSL) were evaluated for the combined delivery of IR-780 iodide photothermal agent and human serum albumin nanoparticles conjugated with paclitaxel (HSA-PTX). IR-780 iodide is a lipophilic dye that was incorporated in the lipid bilayer of the liposomes (CAP-ITSL), while the HAS-PTX nanoparticles were encapsulated in the CAP-ITSL liposomes. The CAF-responsive thermosensitive liposomes developed enhanced drug accumulation to solid tumors and FAP-α-mediated responsiveness. These liposomes effectively promoted FAP-α-mediated targeting of CAFs, which resulted in their FAP-α responsive cleavage and effective release of the HSA nanoparticles for enhanced PTX accumulation at the tumor site. Under the application of an NIR laser irradiation, IR-780 iodide produced local hyperthermia, which is beneficial for deep tissue penetration by expanding the tumor interstitial space; this further promotes drug accumulation and tumor cells apoptosis.

Desmoplastic tumors are the most aggressive type of tumors. They are characterized by a rapid increase in the proliferation of α-SMA positive CAFs and by increased deposition of ECM components. The pancreatic ductal adenocarcinoma (PDAC) is the most characteristic example of tumors establishing a dense heterogenic desmoplastic stroma, maintaining abundant stromal cells and CAFs. The CAFs represent the major source of fibrotic ECM components (collagen, fibronectin, HA) of tumor stroma in PDAC and coordinate the signaling cascades between tumor cells. Thus, CAFs specific targeting is a prominent therapeutic strategy for pancreatic cancer, and it provides great opportunities and challenges. This was shown in a recently review by Liu et al. [142]. In a study by Hou et al. [143], cationic PAMAM polymeric dendrimers delivering doxorubicin via disulfide bonding were designed for CAFs targeting in order to deeply penetrate into desmoplastic tumors of PC-3/CAFs tumor-bearing mice. The PAMAM dendrimers were modified with hyaluronic acid and further cross-linked by a CAP peptide responsive to FAP-α biomarker on CAFs for tumor targeting ability. Upon administration, FAP-α targeting and cleavage promoted the disassembly of the system at the tumor site and the rapid release of doxorubicin and hyaluronic acid that were effectively internalized by CAFs and tumor cells; thus promoting a synergistic antitumor effect. In the tumor tissues, TGF-β, α-SMA, and FAP-α were, importantly, suppressed. This promoted the degradation of tumor fibrotic stroma. The most recent and prominent research field associated to CAFs depletion is presented by the application of DNA vaccines. Through FAP-α targeting, these promote synergistic antitumor and immunity effects that are highly attributed to the upregulation of CD8^+^ T cells and downregulation of macrophages [136,144]. In preclinical studies and recent phase I clinical trials, chimeric antigen receptors (CAR) T cells with FAP-α specificity effectively inhibited FAP^+^ CAFs activity [136,145].

#### 3.3.2. Synergistic CAFs Inactivation with Antitumor Targeting

CAFs inactivation has been investigated in combinational systems with chemotherapeutic drugs and biomolecules for its synergistic antitumor and antimetastatic effect. In a recent study, Huo et al. [146] developed a methotrexate (MTX) nanodrug with a hydroxyethyl chitosan (glycol chitosan, GC) backbone and DEAP side chains (MTX-GC-DEAP) for the co-delivery of quercetin (QUE). The self-assembling nanodrug delivering QUE, which is known for its antifibrotic and antimetastatic effects, expressed mild acid pH sensitivity to TME (pH 6.8) and targeting ability to folic acid receptors (FR) of tumor cells due to MTX. QUE effects originated from inhibiting TGF-β-mediated CAFs activation, through suppression of TGF-β paracrine secretion and of β-catenin/PI3K signaling pathways. QUE downregulated the epithelial-to-mesenchymal transition (EMT), the MMPs secretion and the secretion of certain biomarkers such as α-SMA, collagen production, and pro-metastatic growth factors (VEGF, TGF-β). The evaluation of the multi-responsive system in 4T1 tumor-bearing BALB/c mice demonstrated the inhibition of pro-metastatic initiation by promoting CAFs inactivation and direct regulation on TME as expressed by the suppression of EMT and of blood vessel invasion. In another study, the synergistic effect of the combined delivery of doxorubicin and TGF-β receptor inhibitor was investigated by Zhou et al. [147] using hydroxyethyl starch PLA (HES-PLA) nanoparticles. In mice models of subcutaneous 4T1 tumors, the doxorubicin-loaded HES-PLA nanoparticles demonstrated enhanced inhibition activity on the progression of EMT and increased suppression of tumor growth and metastasis.

Vitamin D receptor (VDR) has been intensely evaluated for CAFs normalization (Figure 8). It plays a key role as a genomic suppressor, which promotes an inactivated state of CAFs [148]. The vitamin D analogues such as calcipotriol or paricalcitol were studied as potential VDR ligands in clinical trials [149]. Moreover, active metabolites of vitamin A were evaluated in clinical trials due to their potency to inhibit aggressive tumor progression [150]. Also, the TGF-β signaling was evaluated for effective CAFs normalization via galunisertib. This is an inhibitor of TGFR-β I kinase, and acts as an immunosuppressor for CAFs inactivation [151]. Currently, CAFs targeting is the focus of interest in cancer nanomedicine research against aggressive tumors with a strong and dense stroma by the application of CAFs targeting peptides for deep penetration and increased accumulation, and inhibitory agents of signaling pathways between CAFs and tumor cells for regulating immunosuppression effects and drug resistance. This is outlined by Meng et al. [152] and by Guo et al. [153]. Recently, gold nanoparticles with known antiangiogenic effects were evaluated by Zhao et al. [154] in colorectal cancer tumor-bearing mice for their suppressing effect on CAFs. Evidently, the gold nanoparticles can reduce the production of stromal collagen type I and inhibit the expression of pro-fibrotic signaling via Akt pathway. This includes the downregulation of CTGF, TGF-β1, and VEGF expression levels.

**Table 4 pharmaceutics-16-00179-t004:** Nanomedicines based on biomaterials targeting stroma CAFs.

Targeting Effects	Carrier Type	Therapeutic Agent	Characteristics	Ref.
CAFs depletion	Anti-FAP-IL liposomes	Single-chain Fv fragments against FAP (scFv’FAP)	Specifically and efficiently respond to FAP-α cell surface biomarker	[139]
	Cleavable amphiphilic peptide (CAP) nanoparticles	Doxorubicin/CAP	Disturbed the stromal barrier and increased drug intratumoral accumulation	[140]
	Thermosensitive liposomes (CAP-TSL)	IR-780 photothermal agent/paclitaxel/human serum albumin	Increased cells apoptosis, expanded tumor interstitial space, promoted deep tumor penetration	[141]
	Poly(amidoamine) (PAMAM) hyaluronic acid nanoparticles	Doxorubicin/CAP peptide	Deep intratumoral penetration, suppression of TGF-β, α-SMA, and FAP-α, degradation of tumor fibrotic stroma	[143]
	Vaccines	FAP targeting	Synergistic antitumor immunity effect	[136,144,145]
Synergistic inactivation	Glycol chitosan–DEAP nanodrug	Methotrexate/quercetin	Inhibition of pre-metastatic initiation, downregulation of metastasis promoting factors inactivation of CAFs	[146]
	Hydroxyethyl starch PLA nanoparticles	Doxorubicin/TGF-β receptor inhibitor	Suppression of tumor growth and metastasis	[147]
	Au nanoparticles	Photodynamic therapy	Inhibit the expression of pro-fibrotic signaling via Akt pathway	[154]

### 3.4. The Tumor Hypoxia

Another therapeutic target of responsive nanomedicine is the TME hypoxia (Figure 9). This is a direct consequence of heterogenic vasculature and fluctuating blood flow, and results in insufficient oxygen diffusion and perfusion within the tumor environment. The rapid proliferation rate of tumor and stromal cells creates excessive consumption of supplied oxygen, nutrients, and energy [155]. Imbalance in the diffusion mechanisms of oxygen supply is observed at depths after 70–150 μm from peripheral tumor blood vessels. This results in gas oxygen (gas-O_2_) levels falling below 1–2% in hypoxic solid tumors. There are two types of hypoxia: the chronic, wherein oxygen’s concentration is characterized by a longitudinal gradient drop for a prolonged time period of several hours; and the acute, in which tumor ECs and stromal cells are attached to vasculature with deprived oxygen perfusion [156]. Extensive research has resulted in the understanding of hypoxia mechanisms and their effects on tumor biology by participating in the regulation of angiogenesis, metastasis, and multidrug resistance (Table 5) [157,158]. Hypoxia-regulated genes are expressed among various tumor types with high hypoxic gene expressions such as the squamous cell carcinoma (SCC) of the head and neck, lung, and cervix tumors; these have also been investigated [159].

#### 3.4.1. GLUT Targeting Nanomedicines

A central factor in targeting TME hypoxia is effective exploitation of transcription factors related to various regulatory pathways of glycolysis, oxygen homeostasis, MDR, and resistance to apoptosis. Such factors within TME include the carbonic anhydrase IX (CA-IX), the glucose transporter-1 and 4 (GLUT-1, 4), and the hypoxia inducible factors (HIF) family (including HIF-1, HIF-2, HIF-3). These act as oxygen regulators to stabilize hypoxic conditions and promote tumor cell survival and angiogenesis through PDGF secretion [160]. Specifically, the blockage of GLUT-1 and GLUT-4 transporters with glucose-modified PLGA and chitosan nanoparticles was selected as an active targeting strategy to limit nutrient supply to tumor cells by Abolhasani et al. [161]. They found that this glucose-modified nanoparticle effectively inhibited GLUTs function and stimulated glucose deprivation and increased apoptotic enzyme expression. In a recent study by Sun et al. [162], the ATRP copolymerization of glucose-containing methacrylate (GluMA) and OEGMA was used for the conjugation of interferon-α (IFN-α) for effective GLUTs targeting. The study outlined the importance of optimizing glucose content. This is because excessive glucose concentration resulted in inhibition of the antitumor activity, while the optimal system showed enhanced tumor targeting and antitumor immunity; this was expressed by the secretion levels of TNF-α and IL-2 cytokines. In general, glycosylation of nanoparticles has been exploited for its increased GLUT-targeting ability, and it promotes metabolic changes and immune responses. This has been reviewed by Torres-Perez et al. [163]. Moreover, glycolysis is the main source of energy production in hypoxic and aerobic tumor sites (the Warburg effect). This demonstrates the importance of nanomedicines targeting the glycolytic mechanisms. In a recent paper by Geng et al. [164], the targeting of glycolytic enzymes and transporters, and combinational strategies based on nanomedicines were reviewed. Also, glucose metabolism in hypoxic tumor sites highly activates cancer stem cell (CSCs) phenotypes, and this promotes the activation of a stem-like polarization that is mainly regulated by the transcription factor HIF-1α and the Notch pathways [165]. Shibuya et al. [166] demonstrated the importance of modulating GLUT-1 transporter in regulating the stem-like phenotype in pancreatic, ovarian, and glioblastoma CSCs. By specifically inhibiting GLUT-1 with WZB117, the self-renewal and tumor-initiating activities of the CSCs were effectively suppressed in animal models upon implantation of CSCs.

Moreover, the energy supply blockage promoted by specific starvation therapeutics is highly associated with hypoxia-targeting nanomedicines. This is caused by exploitation of biological metal ions (e.g., Ca^2+^, Zn^2+^, Cu^2+^) in tumor starvation mechanisms. Among them, Cu^2+^ and Zn^2+^ are the most prevalent components of enzymes, and they play a crucial role in energy metabolism, gene expression, and genomic stability. Yang et al. [167] studied the effect of copper-based ultra-small nanoparticles (Cu_2−x_Se) modified with HIF-1α inhibitor and coated with tumor cell membrane. The nanoparticles were effectively transported through the blood–brain barrier upon focused ultrasound application, and accumulated at glioblastoma tumor sites where they exhibited significant antitumor activity by inhibiting HIF-1α expression and enhancing tumor sensitivity to disulfiram. In another study, Wu et al. [168] examined the effect of zinc imidazole metal-organic particles (ZIF-8) modified with hyaluronic acid for systemic glycolytic energy deprivation. The nanoparticles expressed CD44-targeting mechanism. This led to effective tumor accumulation and cellular endocytosis that promoted the disaggregation of zinc core by hyaluronidase. The latter further triggered a decrease of NAD^+^ and inactivation of GAPDH, and obtained strong glycolysis inhibition. Additionally, it suppressed GLUT1 regulation which promoted energy starvation in B16-F10 tumor-bearing C57BL/6 mice. The energy demand of tumor cells is critically supported by increased glucose production that leads to increased expression levels of GLUT1 and GLUT2, and abundant glycolytic enzymes. The phenotyping of tumor cells with elevated glucose expression levels represents a negative prognostic factor that can be exploited in diagnosis for the effective design of personalized nanomedicines. In this respect, glucose nanosensors have been investigated for application in diagnosis by Nascimento et al. [169]. They studied the effectiveness of nanopipette-based glucose sensors functionalized with glucose oxidase (GOx) to quantify single cell intracellular glucose levels.

#### 3.4.2. Multidrug Resistance Targeting Therapeutics

The targeting of HIF transcription factors will not be broadly discussed here. This is because its role in hypoxia mechanisms and in responsive therapeutic nanomedicine has been extensively reviewed [170,171,172]. A comprehensive mini-review on engineered nanomedicines summarizing recent progress on HIF-1 targeting was presented by Zhang et al. [173]. HIF is highly associated with multidrug resistance in solid tumors. It develops as tumor cells develop a defense mechanism against drugs, and this results in drug efflux and decrease of intracellular drug concentration. MDR is an ultimately complex mechanism related to gene mutations, increased DNA repair ability, and epigenetic alterations involved in the regulation of gene expression such as silencing of tumor suppressor genes and overexpression of oncogenes. In MDR, drug efflux from tumor cells is mediated by efflux transmembrane pumps such as permeability-glycoprotein (P-gp) (also known as multidrug resistance protein 1 (MDR1)), and important transmembrane proteins that are ATP-dependent and regulated by HIF proteins [174]. Certain chemotherapeutic agents that act through mitochondrial ROS formation, e.g., cisplatin (CSP or DDP), can affect MDR either through inducing ROS-mediated cancer cell death by activating apoptotic signaling pathways or through promoting drug resistance and activation of HIF-1 cascade due to elevated ROS production. However, ROS-induced HIF-1 activation also leads to severe damage to the surrounding normal tissues. In a recent study, Zhang et al. [175] investigated the effect of cisplatin on inhibiting ROS induced HIF-1 activation and acquired resistance using chitosan-coated selenium/cisplatin nanoparticles. Chitosan provided long circulation and effective nanoparticles accumulation at the tumor site in cisplatin-resistant A549 tumor-bearing mice. At the tumor, cisplatin promoted HIF-1 expression in an ROS dependent function while the selenium antioxidant effect suppressed ROS formation and inhibited HIF-1 activation. The inhibitory effect of selenium nanoparticles was also confirmed by the downregulation of GCLM, P-gp, MDR2, and HIF-1α protein expression levels. The acquired drug resistance promoted by cisplatin application was examined by Zhang et al. [176] in organosilica-coated cisplatin nanoparticles that co-deliver the HIF-1 inhibitor acriflavine (ACF). The nanoparticles were efficiently accumulated at the tumor site, being susceptible to glutathione triggered biodegradation, and released ACF for inhibiting HIF-1 activity by preventing the formation of HIF-1α/β dimers via HIF-1α binding. The synergistic release of cisplatin resulted in a highly effective system suppressing tumor growth and metastasis. Reversing MDR was also studied by Li et al. [177] through the combined delivery of doxorubicin and the HIF inhibitor PX478 by silk fibroin nanoparticles. The nanoparticles were functionalized with folic acid (FA) for cancer cells targeting. The multi-responsive nanoparticles inhibited HIF gene expression by the synergistic action of FA receptor-mediated endocytosis and PX478 inhibition that effectively downregulated MDR1 expression levels, thus eliminating DOX efflux. Neuropilin-1 (NRP1) is another receptor involved in cellular processes related to MDR in tumor cells. The role of NRP-1 on the loss of therapeutic efficacy of lenvatinib was evaluated by Fernandez-Palanca et al. [178] in relation to hypoxia and modulation of HIF-1α. Mamnoon et al. [179] studied NRP-1 as a molecular target for the delivery of iRGD peptide and doxorubicin by hypoxia-responsive PLA-diazobenzene-PEG polymersomes. Evaluation in tumor-bearing animal models demonstrated the accumulation of the polymersomes at the tumor site and the antitumor effect of their drug cargo.

#### 3.4.3. Increasing Chemo-Sensitivity

On the basis of chemotherapy and radiotherapy resistance, the researched solutions focused on sensitizers and in enhancing TME oxygenation. The sensitizers are, in general, chemical compounds (originally nitro-aromatic ring compounds) that act by elevating the tumor cellular sensitivity to ionizing radiation, and produce free radicals and promote DNA damage [180]. A great drawback of the application of chemical sensitizers was the dose-dependent toxicity and adverse effects related to ROS and RNS expression levels. However, more effective sensitizers with low toxicity were designed by exploiting small molecules (O_2_, NO), macromolecules (proteins, peptides, miRNA, siRNA), and nanomaterials (noble metals, ferrite, heavy metals) that act without distressing ROS expression levels [181]. In tumors, TRPA-1 is a channel plasma membrane protein that promotes extracellular Ca^2+^ influx and represents the most abundant redox sensor upregulated in metastatic cells of solid tumors such as human oral squamous cell carcinoma, colorectal cancer adenocarcinoma, and glioblastoma multiform. The TRPA-1 was evaluated as a ROS sensor, since its upregulation-mediated enhanced ROS-induced Ca^2+^ influx as a result of oxidative stress. This promoted anti-apoptotic signaling pathways [182]. Wang et al. [183] studied TRPA-1 inhibition in order to increase tumor chemosensitivity in tumor-bearing mice. They used hyaluronic acid nanogels functionalized with DSPE-PEG nano-micelles conjugated with a tumor homing penetrating tLyP-1 peptide for co-delivering doxorubicin and TRPA-1 inhibitor (AP-18). The combined effects of HA and penetrating peptide resulted in enhanced intratumoral and intracellular localization becasue the HA nanogels disassembled in irregular fragments releasing peptide conjugated nano-micelles upon hyaluronidase effect in TME. The TRPA-1 inhibitor enhanced tumors sensitivity to DOX by suppressing Ca^2+^ uptake and AKT phosphorylation, and promoting regulation of EMT-related proteins.

A redox enzyme that is highly associated with tumor hypoxia is nitroreductase (NTR). This catalyzes the reduction of nitro compounds in the mitochondria by using NADPH. Jang et al. [184] examined the effect of NTR in self-assembled nanoparticles of glycol chitosan loaded with doxorubicin, functionalized with folic acid andmodified with 4-nitrobenzyl chloroformate (4NC). The nanoparticles effectively accumulated at the tumor site due to the combined effects of CS and FA targeting, and disassembled due to the reduction of the chitosan bonded 4NC under hypoxic conditions by the NTR/NADPH cascade. This efficiently promoted DOX antitumor activity. Another study exploiting the hypoxia-mediated bond reduction was by Luo et al. [185]. They examined self-assembling thioether linked dihydroartemicin (DHA) prodrug nanoparticles loaded with chlorin e6 and further encapsulated in core-shell amphiphilic carboxymethyl chitosan polymer particles, grafted with maleimide and 2-nitroimidazole (NI) groups. The nanoparticles showed significantly long circulation time and accumulation ability through the combined chitosan and EPR effect. Once PDT was applied, oxygen consumption was increased and the NI groups were reduced. This destabilized the structure of the chitosan particles, and further promoted drug release at the tumor site. The synergistic effect resulted in suppression of HIF-1α and VEGF expression levels, inhibition of tumor metastasis, and an improved therapeutic effect in LLC tumor-bearing rats and mice.

#### 3.4.4. Targeting M1/M2 Macrophage Polarization

In TME, hypoxia is responsible for inducing the reprogramming of pro-inflammatory M1 macrophages to take a M2 anti-inflammatory phenotype; this promotes tumor aggressiveness and MDR. M1 macrophages play critical roles in innate host immunity and tumor cell death through the production of ROS/RNS and pro-inflammatory cytokines such as IL-1β, IL-6, and TNF-α. M2 macrophages polarization induced by Th2 cytokines such as IL-4, IL-10, and IL-13 play a critical immunosuppressive role in immune responses, under the activation of immune complexes (IC) and TLR ligands, producing anti-inflammatory cytokines such as IL-10, IL-13, and TGF-β to promote tumor development. By altering macrophage responses, hypoxia downregulates the expression of antigens and the release of pro-angiogenetic factors. This exerts immunosuppressive effects that promote cell proliferation. While the M1/M2 macrophage states have been identified, the regulation of macrophage phenotypes remains a complex mechanism. The production of distinct functional phenotypes in macrophages (polarization), as a consequence of hypoxia, is essentially regulated by HIF-1α, which plays a key role in the expression of several genes [186]. Among them, the PTGS gene encoding cyclooxygenase-2 (COX-2) is of major importance, since it is related to increased accumulation of pro-inflammatory signals, and represents an indicator associating inflammation with cancer [187]. COX-2 is secreted by CAFs, M2 macrophages and TME cells inducing cancer stem-like cells activity, MDR, and metastasis. COX-2 induced HIF-1α activity is responsible for biosynthesis of prostanoids, like prostaglandin E2 (PGE2); this is overexpressed in a variety of cancers including breast cancer, osteosarcoma, and colorectal cancer. The HIF-1α/COX2 signaling axis is highly related to resistance, invasion, and metastasis [188]. Karpisheh et al. [189] studied the effect of HIF-1α silencing siRNA and PGE2 receptor antagonist (EP4-antagonist) as a combinational therapy with hyperthermia in tumor-bearing BALB/c nude mice. For this purpose, superparamagnetic iron oxide nanoparticles (SPIONs) functionalized with hyaluronic acid and trimethyl chitosan were used for the delivery of siRNA and the EP4 antagonist. The synergistic effect of hyperthermia, CS and HA resulted in increased SPIONs biodistribution at the tumor site, and effectively delivered the siRNA. The combined effect of HIF-1α siRNA and the EP4 antagonist resulted in (i) suppression of proliferation and migration of tumor cells by decreasing the expression levels of ki-67 gene, anti-apoptotic protein Bcl-2, and MMP-2/-9; and (ii) inhibition of VEGF, FGF, and TGF-β. This promoted suppression of angiogenesis and inhibition of tumor growth.

#### 3.4.5. Combinational Targeting for Increasing Tumor Oxygenation

In order to increase tumor oxygenation, various strategies have been investigated. These include: (i) modulation of tumor blood flow with compounds, such as noradrenaline, benzyl nicotinate, nicotinamide, and pentoxifylline; (ii) high oxygen breathing through hyperbaric chambers, and carbogen breathing; (iii) targeting of tumor vasculature with anti-angiogenetic therapies and vascular disruptive agents [190]. Hyperbaric oxygen (HBO) treatment was used to suppress hypoxia by effectively elevating the oxygen concentration in plasma and, subsequently, enhancing oxygen delivery in tumor tissues independently of hemoglobin. This resulted in reduced tumor growth in breast cancer; however, cervical and bladder cancers appear to be insensitive to HBO [191]. In a recent study by Wang et al. [192], HBO was applied in combination with Abraxane and gemcitabine (GEM) to trigger antitumor activity against murine PDAC tumors, expressing inhibitory activity over ECM by decreasing fibril deposition of collagen I and fibronectin. As CAFs are the main cellular components of ECM, HBO significantly inhibited their activity by suppressing tumor hypoxia that, in combination with abraxane and GEM, resulted in inhibition of CSCs. This is evidenced by the decreased expression levels of CD133 and Sox2 CSCs-related biomarkers, and this further promoted the antitumor activity of the drugs in primary and in metastatic PANC02 tumors.

In a recent review by Wu et al. [65], nanoparticle-based systems targeting TME were presented. These include hypoxia-responsive nanomedicines that effectively target hypoxic TME to promote tumor oxygenation. Tumor oxygenation in order to inhibit hypoxia was examined by Song et al. [193], with the application of manganese dioxide nanoparticles (MnO_2_) coated with hyaluronic acid and modified with mannan-PE ligands in combination with priming of pro-inflammatory M1 macrophages phenotype. The nanoparticles caused elevation of tumor oxygenation. This is shown by the suppression of HIF-1α and VEGF expression levels, and expressed immune-toxicological effects in reprogramming the antitumor M1 phenotype. Furthermore, the nanoparticles had synergistic effect upon administration with doxorubicin in tumor-bearing mice; in which they inhibited tumor growth and cell proliferation. Importantly, the MnO_2_ nanoparticles express a redox-active catalytic behavior toward hydrogen peroxide (H_2_O_2_) to produce oxygen and regulate acidic pH. This is because their Mn^2+^ decomposing products are excellent T1-shortening MRI agents; this is described by Lin et al. [194] in human serum albumin MnO_2_ nanoparticles conjugated with Ce6 photosensitizer. The evaluation of the nanoparticles in tumor-bearing mice showed increased tumor-targeting ability and accumulation efficacy, with elevated oxygenation levels. Additionally, treatment with PDT-enhanced cellular apoptosis, resulted in a large tumor necrosis area. Another effective system promoting oxygenation was studied by Jiang et al. [195], who used MnO_2_ albumin nanoparticles delivering indocyanine green (ICG), a hydrophilic anion drug, for effective PDT of hypoxic tumors. The nanoparticles were evaluated in both CT26 and B16F10 tumor-bearing mice and showed enhanced tumor accumulation efficacy and successful responsive release of ICG in the presence of hydrogen peroxide. The combined effect with MnO_2_ and PDT resulted in enhanced oxygen production and attenuation of hypoxic TME, while elevated distribution of CD3^+^ and CD8^+^ T cells was promoted at tumor sites. In advanced hepatocellular carcinoma (HCC) the anti-angiogenetic agent sorafenib is a first-line treatment. However, the hypoxic TME of advanced HCC regulated anti-apoptotic signaling pathways and promoted immunosuppressive reprogramming, resulting in MDR against sorafenib. Ren et al. [196] studied the synergistic effect of tumor oxygenation and PDT by a strategic hypoxia relieving nanodrug (SHRN). SHRN was based on pegylated liposomes encapsulating MnO_2_-BSA nanoparticles, the oxygen consumption inhibitor atovaquone (ATO), and the photosensitizer hypericin (HY). In this system, oxygen production was effectively increased by MnO_2_ nanoparticles and the synergistic action of ATO with mitochondrial complex III resulted in the blockage of aerobic respiratory chain and the suppression of oxygen consumption. By suppressing hypoxia in TME of tumor-bearing mice, PDT application essentially promoted HY action to react with oxygen and promote increased antitumor effect. The antitumor activity of PDT originated from the elevated generation of ROS as an outcome of electron transfer mechanisms of the photosensitizer and the molecular oxygen intracellularly. A review of the mechanisms associated with PDT efficacy and the limitations induced by hypoxia has been authored by Zhou et al. [197]. In another study, Chang et al. [198] investigated the effect of lipid-PLGA particles co-delivering MnO_2_ nanoparticles and sorafenib as an effective approach for HCC. The nanoparticles promoted oxygen production in orthotopic HCC tumor mice xenografts, and this resulted in suppression of hypoxia and of MDR to sorafenib. As a result, inhibition of tumor cells proliferation and suppression of angiogenesis and metastasis were observed.

In addition to MnO_2_, iron oxide nanoparticles were also studied as Fenton catalysts triggering reduction of hydrogen peroxide to oxygen by ferrous ions. He et al. [199], described the in vitro and in vivo effectiveness of solid lipid calcium dioxide (CaO_2_) nanocarriers (SLNs) for co-delivery of doxorubicin and iron-oleate. At the tumor site, the SLNs dissociated by lipase overexpressed in cancer cells, enabling the release of iron-oleate and CaO_2_ particles. These, in response to acidic cellular environment, released DOX and produced hydrogen peroxide molecules. The Fe^3+^ ferrous ions released from iron-oleate reacted with H_2_O_2_ molecules to produce oxygen and the Fe^2+^ ions created hydroxyl radicals for antitumor chemodynamic therapy (CDT). The latter can induce oxidative damage to tumor tissues through Fenton or Fenton-like reactions of metal catalysts. In another interesting study by Ou et al. [200], the antitumor effect of CDT based on Fenton reaction was exploited. Βlack phosphorus nanosheets (BPNS) functionalized with active photosynthetic Chlorophyceae (Chl) cells and Fe^3+^ ions were synthesized. In this smart system, Chl cells exploited their inherent photosynthetic ability to produce O_2_ and metabolites that, in combination with the BPNS, improved ^1^O_2_ and O_2_ generation. The presence of Fe^3+^ ions resulted in the simultaneous consumption of glutathione and created hydroxyl radicals (·OH) through the reaction with hydrogen peroxide. The ultimate goal was the synergistic PDT/CDT and immune response, since Chl cells stimulate the proliferation and maturation of dendritic cells. Recently, Dong et al. [201] prepared Zn/Cu responsive nanoparticles with improved blood circulation time and increased tumor accumulation in animal models through the EPR effect. The pH-responsiveness of Cu/Zn nanoparticles originated from the ZnO particles which, at acidic pH, dissolve to Zn^2+^ ions. Enhanced CDT efficacy was observed at the acidic tumor pH, since Cu/Zn nanoparticles dissolved to Cu^2+^ ions that were further reduced by glutathione to Cu^+^. This effectively generates hydroxyl radical from hydrogen peroxide through a Fenton-like reaction. Moreover, CaO_2_ nanoparticles coated with a pH-sensitive methacrylate based copolymer were studied by Sheng et al. [202] to enable tumor oxygen generation and improved PDT therapy in MIA PaCa-2 tumor-bearing mice. The combination of CDT with immunotherapy was examined by Zhang et al. [203]. They applied liposome nanoparticles carrying copper-oleate and the HIF-1 inhibitor acriflavine (ACF), for combined antitumor immune responses. The liposomes dissociated in the acidic hypoxic TME and the released copper ions catalytically reduced hydrogen peroxide to highly active hydroxyl radicals. The activity of copper ions was supported by ACF that effectively inhibited the HIF-1/glutathione pathway. This suppressed the expression of programmed death ligand-1 (PD-L1), reduced the extracellular expression levels of lactate and adenosine, and promoted immunogenic cell death (ICD).

#### 3.4.6. Synergistic Targeting with Antioxidants

The synergistic effect of hypoxia-targeting nanomedicines in combination with chemotherapeutic agents and antioxidants has been extensively investigated, due to their effects in reoxygenation and ROS expression levels [204,205,206]. The overproduction of ROS can promote oxidative stress (OS) and induce oxidative damage of biomolecules as DNA, lipids and proteins. This eventually promotes the death of normal cells while in tumor tissues, due to increased energy demand and metabolic modifications, the demand on ROS production is excessively increased. The reduction in antioxidant level and/or the disruption of redox equilibrium within TME can further promote tumor cell growth and progression. Among various antioxidants studied in tumor therapeutics, polyphenols are associated with cancer cell apoptosis, inhibition of proliferation, and downregulation of COX-2 and tumor gene expression. Vitamins and minerals elicit their antioxidant action by maintaining DNA methylation inhibiting cancer cell proliferation and progression [207]. Vitamins such as Vitamin C (ascorbic acid, AA) are capable of scavenging and neutralizing tumor-generated ROS, providing normalization of OS within local tumor sites. This interaction promotes the production of dehydroascorbic acid (DHAA) that is related with GLUT-1,3, and 4 transporters for effective and rapid cell influx. Intracellularly, DHAA is converted into AA depleting glutathione and ATP enzymes. This affects the expression levels of hypoxia signaling regulation factors [208]. However, the therapeutic exploitation of AA is limited by the ultra-high doses that are required for efficacy and by its chemical instability [209]. Thus, palmitoyl ascorbate (PA; an acylated derivative of AA) also features antitumor activities, and has been incorporated in ROS-scavenging nanomedicines. Sawant et al. [208] investigated the effect of palmitoyl ascorbate PEGylated liposomes (PEG-PAL) in vitro and in vivo in BALB/c mice bearing 4T1 tumors. PEG-PAL liposomes exhibited enhanced effectiveness in suppressing tumor growth compared to free ascorbic acid. The mechanism of action of PEG-PAL was similar to that of ascorbic acid, since liposomes acted as ROS scavengers inhibiting extracellular ROS. In another study, Yang et al. [209] examined the combinational delivery of PA and doxorubicin by liposomes for efficient synergistic effect in suppressing tumor growth. This is because PA has a similar mechanism of action with DOX associated with reduced cardiotoxicity without delaying DOX activity. The evaluation in BALB/c mice and SD rats showed the synergistic antitumor effect of the PA-DOX liposomes, which caused an increased expression of tumor apoptotic cells and a suppression of Ki-67 and CD31 protein expression levels. 

**Table 5 pharmaceutics-16-00179-t005:** Nanomedicines based on biomaterials targeting tumor hypoxia.

Targeting Effects	Carrier Type	Therapeutic Agent	Characteristics	Ref.
GLUT	PLGA-chitosan particles	GLUT-1	Glucose deprivation, increased apoptotic enzymes expression	[161]
	Glucose-Methacrylate-OEGMA nanoparticles	Interferon-α	Tumor targeting and antitumor immunity	[162]
	Cu particles/tumor cell membrane coating	HIF-1α inhibitor/disulfiram	Enhanced tumor sensitivity	[167]
	Zn-imidazole–hyaluronic acid particles	DNAzymes	Antitumor effects inhibiting glucose energy	[168]
	Nanopipette sensors	Glucose Oxidase	Identification of intracellular glucose level	[169]
Multidrug Resistance	Se/chitosan nanoparticles	Cisplatin	Suppressed ROS formation, inhibited HIF-1α, MDR-2, P-gp	[175]
	Organosilica particles	Cisplatin/Acriflavine	Inhibition of tumor growth and metastasis	[176]
	Silk fibroin particles	Doxorubicin/PX478 HIF inhibitor	Downregulation of MDR1 and P-gp	[177]
	PLA-diazobenzene-PEG polymersomes	iRGD peptide/Doxorubicin	Increased accumulation, inhibition of tumor growth	[179]
Chemo-Sensitivity	Hyaluronic acid nanogels/DSPE-PEG nano-micelles	Doxorubicin/TRPA-1 inhibitor	Enhanced tumors sensitivity, antitumor and antimetastatic effects	[183]
	Chitosan-FA particles	Nitroreductase/Doxorubicin	Hypoxia triggered effective antitumor action	[184]
	CM-chitosan-maleimide particles	Dihydroartemicin/PDT	Suppression of HIF-1α and VEGF, inhibition of tumor metastasis	[185]
M1/M2 polarization	Iron oxide-hyaluronic acid-chitosan nanoparticle	HIF-1α siRNA/PGE2 receptor antagonist	Suppression of proliferation, migration, angiogenesis, decreased protein levels	[189]
Combinational	MnO_2_–hyaluronic acid nanoparticles	Doxorubicin	Inhibiting tumor growth and cell proliferation	[193]
	Human serum albumin MnO_2_ nanoparticles	Chlorin e6/PDT	Tumor targeting ability, increased accumulation, elevated oxygen levels, tumor necrosis and apoptosis	[194]
	MnO_2_–albumin nanoparticles	Indocyanine green/PDT	Enhanced oxygen production, antitumor effect	[195]
	DSPE-PEG liposomes/MnO_2_-BSA nanoparticles	Atovaquone/hypericin/PDT	Suppressing hypoxia, increased antitumor effect	[196]
	Lipid-PLGA-MnO_2_ particles	Sorafenib	Hypoxia suppression, inhibited tumor cells proliferation, suppressed angiogenesis and metastasis	[198]
	Solid lipid calcium peroxide (CaO_2_) nanocarriers	Doxorubicin/iron-oleate/Chemodynamic theapy	Oxidative damage to tumor tissues	[199]
	pH-sensitive methacrylate–CaO_2_ particles	CaO_2_ particles/PDT	Increased tumor oxygenation	[200]
	Liposome nanoparticles	Cu-oleate/Acriflavine	Immunogenic cell death, combined antitumor immune responses	[201]
Antioxidants	PEGylated liposomes	Palmitoyl ascorbate	Suppressed tumor growth	[206]
	Liposomes	Doxorubicin/Palmitoyl ascorbate	Suppressed tumor growth	[207]

### 3.5. The Tumor Acidosis

Tumor and stromal cells use aerobic glycolysis for their amplified energy supply requirements, as a direct consequence of hypoxia and defective vasculature. Aerobic glycolysis is an oxygen-independent process known as the Warburg effect. However, even in normally oxygenated tumor regions, the main energy supplier remains aerobic glycolysis in about 80% of solid tumors [210]. In aerobic glycolysis, glucose constitutes the main macronutrient of tumor cells for their biosynthetic requirements and follows the lactate metabolic pathway, through GLUT transporters producing amplified levels of lactic acid (lactate). The main transcriptional factors of glycolytic activity regulating lactate production are the HIF-1α and c-Myc regulatory genes [211] that promote the overexpression of varied glycolytic enzymes such as lactate dehydrogenase A (LDHA), and monocarboxylate transporters (MCTs) such as MCT1 and MCT4 [212]. Mainly, the upregulation of LDHA gene favors the activity of LDH-5 and inhibits the activity of LDH-1, and promotes the conversion of pyruvate to lactate. Through this metabolic pathway, elevated amounts of lactate, protons (H^+^), and carbon dioxide (CO_2_) are secreted into TME. This leads to acidosis [213]. Acidosis regulates the metabolism of innate and adaptive immune cells by: (i) hindering the function of CD8^+^ T, natural killer (NK), natural killer T (NKT) and dendritic (DC) cells; (ii) supporting regulation of FOXP3^+^ T cells (Treg); and (iii) promoting M2 activated macrophage polarization. Overall, the acidic TME is an immunosuppressive incubator of pro-oncogenic and tumorigenic factors, and has been extensively studied for targeted nanomedicine applications [214,215,216]. The glycolytic metabolic pathway and acidic pH gradient are key participating factors in MDR due to their activation of enzymes and proteins responsible for resistance, efflux of drugs through P-gp, and stimulation of migration [217,218].

In tumors, a unique pH-gradient effect is established, with extracellular pH levels (pH_e_) being more acidic (6.4–7.0) and intracellular pH (pH_i_) being more alkaline (7.25–7.50) (Figure 10). Distinct pH variations exist in tumor cell organelles, and they can be divided into acidic, such as nucleosomes and lysosomes with a pH of 5.5 and 5.0, respectively; or alkaline, such as mitochondria and cytoplasm, which have a corresponding pH of 8.0 and 7.2, respectively. The pH gradient is associated with the expression of membrane transporters such as MCT1, MCT4, carbonic anhydrases, and sodium-bicarbonate co-transporter (NBC). These participate in the translocation of lactic acid, CO_2,_ and its bicarbonate ion byproducts. Other mechanisms influencing TME acidity are the efflux of endosomes acidic cargo and the release of the acidic intracellular comportments of necrotic cells. In stimuli responsive nanomedicine (Table 6), pH sensitivity has been highly exploited and reviewed [219,220,221,222,223]. Apart from drug delivery systems, tumor acidosis was targeted by pH-regulating molecular systems at various stages of clinical trials (this is described by Corbet et al.) [224], and by TME sensitive platforms for combined endogenous stimuli responsive effects (as reviewed by Wang et al. [225]). 

#### 3.5.1. pH-Sensitive Peptides in Acidic Tumor Targeting

A strategy for exploiting TME acidity is based on pH responsive peptides which under physiological conditions interact weakly with the cellular membrane but at TME create stable transmembrane complexes promoting nanomedicines internalization. Yadav et al. [226] examined chitosan nanoparticles modified with a pH-sensitive cRGD peptide (RGD-CHNP) for the delivery of raloxifene (Rlx) in NOD/SCID 4T1 tumor-bearing mice. The nanoparticles presented enhanced tumor accumulation by RGD peptide active targeting in α_v_β_3_ integrin expressing breast cancer cells and expressed enhanced antitumor effect, inhibiting angiogenesis and migration by suppressing the regulation of osteopontin (OPN), thus inhibiting Akt and ERK signaling cascade. The combination of receptor-mediated specific binding and acidic pH was exploited by Han et al. [227]. They designed glycogen nanoparticles functionalized with doxorubicin via a pH responsive hydrazine-based bond and β-galactose, with selective binding affinity to the asialoglycoprotein transmembrane receptor (ASGPR) on hepatic cancer cells. Upon ASGPR binding, cellular internalization and degradation of the nanoparticles was triggered and pH-sensitive DOX release was promoted. The nanoparticles were evaluated in BALB/c nude, hepatic tumor-bearing mice. They exhibited enhanced accumulation at the tumor site and efficient antitumor activity of DOX inhibiting tumor growth. Palanikumar et al. [228] studied PLGA nanoparticles cross-linked with bovine serum albumin (BSA) and conjugated with pH-responsive membrane peptide (ATRAM) for the delivery of doxorubicin attached to triphenylphosphonium (TPP) in tumor-bearing C3H/HeJ mice. BSA provided long circulation time of the nanoparticles, and this resulted in effective intracellular localization in response to acidic pH, owing to the ATRAM peptide. The BSA coating was susceptible to GSH-mediated degradation that promoted the controlled release of DOX-TPP and resulted in enhanced mitochondria DOX accumulation. This effectively inhibited tumor volume and mass while exhibiting no apparent toxicity to healthy tissues.

Among pH-sensitive nanomedicines for tumor therapy, nanogels have been extensively investigated owing to their unique characteristics. These include self-assembly ability, stability upon systemic circulation, improved drug delivery compared to polymeric nanoparticles, high specificity and tissue penetration through EPR due to their small size, and bioconjugation activity for microenvironment responsive therapeutics [229]. Biomaterials such as hyaluronic acid, chitosan, DNA, and alginate were evaluated for tumor targeting nanogels with pH sensitivity that was due to pH-responsive peptides or pH-sensitive degradation of the cross-linked drugs and molecules. Ding et al. [230] studied hyaluronic acid nanogels cross-linked with pH-sensitive E3 (GY(EIAALEK)3GC) and K3 (GY-(KIAALKE)3GC) peptides (HA-cNCs) for targeted delivery of cytochrome C (CC) and saporin proteins to CD44 overexpressing MCF-7 breast cancer cells. The intracellular localization of the nanogels was promoted by CD44 receptor-mediated endocytosis due to HA, which triggered the endosomal degradation of the E3/K3 pH-sensitive cross-linked peptides and the release of the loaded proteins. The triggered release of CC and saporin from the nanogels resulted in a combined antitumor effect against breast cancer cells. CC is a hemeprotein weakly connected in the inner mitochondrial membrane, that participates in ATP synthesis. During the early apoptotic phase, detachment of CC is stimulated by ROS production. This leads to CC efflux into the cytosol, and acts as a regulator of apoptotic stimuli in cancer cells. Moreover, saporin is a ribosome-inactivating protein involved in the inhibition of protein synthesis in the cytosol; this results in cell death.

#### 3.5.2. Metals and Metal Oxides in Acidic Tumor Targeting

Metal oxide nanoparticles have attracted research interest in emerging tumor therapeutic and diagnostic applications. Investigation of these nanoparticles has expanded on varied strategies including conjugation, combination with radiotherapy or chemotherapy, and activity based on external or internal stimuli. Several research approaches that combine the effects of metal oxides (MO) with targeting the acidic TME were developed in order to obtain enhanced antitumor efficacy. The interest on MO nanoparticles is due to their pro-apoptotic activity; inhibition of tumor cell growth and metastasis, and ROS production [231,232]. A characteristic example of an MO is cerium oxide nanoparticles (nanoceria). These are inorganic antioxidants that at physiological pH express catalytic mimicking activity to quench ROS effect. At acidic pH, however, they function as oxidases and increase oxidative stress and apoptosis. Gao et al. [233] studied multi-responsive nanoceria particles for the delivery of doxorubicin. The particles were coated with glycol chitosan and were bestowed with tumor-targeting ability by CXCR4 antagonist (AMD11070). An important axis connecting tumor cells and TME is the CXCR4/CXCL12 signaling, based on the CXC G protein-coupled chemokine receptor 4 (CXCR4 or CD184) that is overexpressed in various human tumors including human retinoblastoma. CXC chemokine ligand 12 (CXCL12, or stromal-derived-factor-1, SDF-1) is a ligand that acts through binding to the CXCR4 and promotes cancer stem cell phenotype, tumor progression, invasion, and metastasis. The nanoceria particles were evaluated for their antitumor activity on retinoblastoma cells. They expressed elevated internalization that significantly increased ROS production at acidic pH. This resulted in the inhibition of tumor growth, and there was substantial tumor size suppression and reduction in blood vessel leakages, in orthotopic models of genetic p107s mice.

Manganese dioxide nanoparticles (MnO_2_) represent promising theranostic candidates, combining TME oxygenation triggered by MnO_2_ reduction effect on ROS, with photodynamic therapy and pH-responsiveness. Yang et al. [234] studied hollow MnO_2_ nanoparticles functionalized with PEG for the combined delivery of doxorubicin and the photodynamic agent Ce6. At the acidic tumor pH, the degradation of MnO_2_ nanoparticles was promoted by reaction with protons and GSH. This generated Mn^2+^ ions and led to the oxygenation of the tumors and the combined release of DOX and Ce6. This further promoted the inhibition of tumor growth. Antitumor immune responses were also observed. For example, there was significantly decreased population of M2 macrophages and suppressed expression levels of IL-10. Tumor acidosis was exploited as an endogenous stimulus by Chen et al. [235] for the targeting effect of FA-conjugated MnO_2_-coated mesoporous silicon nanoparticles. The nanoparticles were loaded with metformin (Me), an oral drug for type 2 diabetes, and fluvastatin sodium (Flu), an inhibitor of monocarboxylate transporter 4 (MCT4 protein) that is responsible for mediating the intracellular lactate/H^+^ efflux. The nanoparticles expressed effective targeting affinity to folate receptor for enhanced internalization and intracellular degradation of the MnO_2_ particles by GSH, through oxidation reduction; this resulted in the release of Me and Flu. The synergistic effect of the drugs successfully regulated the pyruvate metabolic pathway, and promoted the production of elevated lactate levels and suppression of the lactate efflux. This further induced intracellular acidosis that promoted tumor cell death, suppressed tumor growth, and inhibited metastasis in MCF-7 tumor-bearing nude mice.

Gold nanostructures are highly applied in tumor targeting, since upon internalization by tumor cells they act as sensitizers to radiation therapy. The advantages of gold nanoparticles include efficient transportation through the leaky tumor vasculature, surface modification by thiol linkages, and use in clinical applications. Rauta et al. [236] studied the conjugation of gold nanorods with charge-reversal poly(Glu-co-Lys) polypeptides with pH responsiveness. They effectively switched charge at the acidic extracellular TME, and this enabled their internalization in tumor cells. The evaluation of the Au nanorods in orthotopic pancreatic tumors resulted in enhanced accumulation at the tumors’ periphery and the hypoxic core of large tumors. No abnormalities were observed in normal organs and there were no hematological deviations; this indicates the safety of the gold nanorods. Another example of charge-reversal responsive polymers induced by pH acidity was studied by Xue et al. [237] in doxorubicin-loaded superparamagnetic iron oxide nanoparticles (SPIONs) modified with citraconic anhydride-dextran (Dex-COOH) and cystamine-dextran (Dex-SS-NH_2_). The nanoparticles showed a pH-responsive negative charge decline due to the acid-sensitive dextran coating; this enabled the internalization of the nanoparticles and the lysosomal escape by switching the charge from negative to positive. Subsequently, the nanoparticles—due to the presence of the disulfide bond—decomposed under the effect of GSH, and this triggered DOX release that promoted antitumor activity. This is shown by the significant inhibition of tumor volume in CT26 tumor-bearing mice. Effective accumulation of the nanoparticles in tumor tissue was observed with low non-specific tissue toxicity. In a study by Angelopoulou et al. [238], SPIONs functionalized with PMAA-g-PEGMA polymers and conjugated with canagliflozin via pH-sensitive bond were evaluated in PDV C57 tumor-bearing mice for their antitumor effect. Canagliflozin is a type 2 diabetes drug that acts through inhibition of sodium-glucose transporter protein (SGLT2), and takes advantage of the TME hypoxia. The nanoparticles expressed enhanced tumor accumulation by the application of a static magnetic field gradient and the pH-sensitive canagliflozin release was triggered. This provided efficient antitumor activity that, in combination with radiotherapy, inhibited tumor growth at a significantly higher degree compared to either monotherapy (drug or radiotherapy).

#### 3.5.3. Biomaterial Based Polymeric Nanomedicines in Acidic Tumor Targeting

Another highly investigated and widely reviewed type of nanomedicine is polymeric systems combined with biomaterials for pH-responsive TME targeting. These include hydrogels [239], polymer nanoparticles [240,241], and micelles [242]. Despite the effort, the complex biological characteristics and aggressiveness of the acidic microenvironment of solid tumors remains a challenge for effective delivery. Since TME acidosis is not considered a limiting barrier, but signifies a micromilieu for smart targeted drug delivery, promising polymeric nanomedicine strategies were studied. Among them, hydrogels are injectable systems for in situ administration of drugs that enable the localized application at tumor site and can be designed in order to acquire pH-stimuli responsiveness and self-healing properties. N-carboxyethyl chitosan (CEC) hydrogels cross-linked with dibenzaldehyde-terminated poly(ethylene glycol) (PEGDA) and conjugated with doxorubicin were injected upon subcutaneous injection in hepatocellular liver carcinoma-bearing rats, to be evaluated for their antitumor activity. The hydrogels effectively accumulated at the tumor site and pH-responsive DOX release was triggered. Moreover, the hydrogel promoted self-healing activities due to the Schiff-base linkage between CEC and PEGDA [243].

In another study, Megahed et al. [244] evaluated pH-sensitive PEGylated chitosan niosomes for the delivery of Tamoxifen (Tam); this is a hormone antagonist used in breast cancer therapy. Chitosan was used as a pH-sensitive polymer and PEG provided the necessary long-circulation properties. Tam is a selective estrogen receptor modulator (SERM) with the activity of binding to estrogen receptors and promoting agonist or antagonist effects, depending on the targeted tissue. Tam represents a promising treatment for estrogen receptor-positive (ER^+^) breast cancer and for stromal targeting of pancreatic ductal adenocarcinoma (PDAC). The evaluation of cell cycle analysis revealed that the presence of chitosan and PEG in niosomes had a great influence on the induced apoptosis. Chitosan promoted apoptosis over necrosis of tumor cells, while PEG presence increased apoptotic and necrotic populations. The evaluation of the niosomes in breast tumor-bearing rats showed elevated antitumor efficacy and increased Tam accumulation at the tumor site. Chitosan is preferentially applied in tumor acidosis, because its abundant amino groups on the polysaccharide chain obtain a positive charge under acidic pH. Thus, an innate pH-responsiveness is generated by chitosan, and this enables its application in screening even for deep analysis of invasive cancer cells. This was reported by Ivanova et al. [245]. Chitosan micro-sized particles were evaluated for screening of tumor progression, in response to acquired resistance of the acidic TME. Toxicity was hypothesized to be associated with biological and chemical metabolic changes of acidic microenvironment and pH gradient effect. The highly invasive metastatic tumor cells have a strong negative charge, and thus they electrostatically attach to the chitosan micro-formulation, and this enables the screening of tumor metastasis. 

Stimuli pH-responsive polymeric nanoparticles are the focus of interest in a wide range of cancer-targeting applications. Relative research includes engineered nanoparticles that are able to respond to TME endogenous stimuli [246]; pH-responsive activity is based on charge-shifting polymer structures, acid labile linkages, and pH-responsive cross-linkers [240]. Zhao et al. [247] studied cross-linked polymeric nanoparticles with folic acid (FA) and galactose (GAL) targeting activity and dual pH/redox-sensitivity due to the PDPA and PDEMA cross-linked block copolymers, respectively. The amphiphilic cross-linked polymers formed self-assembled nanoparticles loaded with doxorubicin and were evaluated in HepG2 hepatocellular carcinoma cells. GAL was responsible for selectively binding to asialoglycoprotein (ASGPR) receptors of HepG2 cells and FA to folate receptors, and promoting dual active targeting for efficient internalization. Due to the protonation of the tertiary amine at acidic pH and the reduction of the disulfide bond by GSH, increased DOX release was promoted intracellularly; this resulted in increased cytotoxicity and apoptosis. Another example of charge-shifting polymers was reported by Yuan et al. [248], who studied zwitterionic polymers based on block copolymers of PCL-*b*-PAEP. The copolymers were composed of equal anion and cation groups on their backbone chain, giving them high hydrophilicity that promotes resistance to protein adsorption, avoidance of rapid recognition by immune system, and delayed blood clearance; these, therefore, represent dynamic alternatives to PEG. The PCL-*b*-PAEP block copolymers were further grafted with thiol derivatives of cysteamine hydrochloride and TMA, resulting in positively charged polymers that were further reacted with 2,3-dimethylmaleic anhydride to acquire pH-sensitivity. The polymers were self-assembled in micelles encapsulating doxorubicin with surface charge-switching ability in response to the acidic TME. The evaluation of the micelles in MDA-MB-231 tumor-bearing mice, provided evidence for enhanced tumor cell internalization and inhibition of tumor growth. Wang et al. [249] investigated charge-shifting PDPA polymers in micelle-type nanoparticles which were loaded with iron oxide nanoparticles (IONPs) and *β*-lapachone (La). The pH-responsive PDPA-modified IONPs were further incorporated in H_2_O_2_-responsive polymeric prodrugs of PEG-polycamptothecin. Thus, dually responsive nanoparticles were obtained, and they expressed pH and H_2_O_2_ sensitivity. This resulted in acidic-mediated degradation in the endosome/lysosome environments due to the shifting pH-responsiveness of PDPA. Thus, La was released and catalyzed by nicotinamide adenine dinucleotide (phosphate): quinone oxidoreductase 1 (NAD(P)H: NQO1), and this produced elevated levels of hydrogen peroxide. Then the newly produced H_2_O_2_ reacted with iron ions to further promote the generation of toxic ROS levels with elevated expression of hydrogen peroxide species promoting the degradation of the peroxalate ester linkages. Thus, the release of camptothecin was triggered. The synergistic effect of the nanoparticles resulted in effective antitumor activity in A549 tumor-bearing mice, and this significantly inhibited tumor volume and tumor growth (IRG) and caused low systemic toxicity. 

PLGA nanoparticles have been highly evaluated in nanomedicine, including pH-responsive applications, owing to excellent biocompatibility, biodegradability, and ease of functionalization properties. Liang et al. [250] studied PLGA nanoparticles coated with BSA and encapsulating doxorubicin and graphene quantum dots (GQDs). GQDs have fluorescence properties for cellular imaging. The pH-responsive DOX release was accomplished by the biodegradation of the PLGA structure and the protonation of daunosamine group in the acidic environment. In another study by Meng et al. [251], PLGA nanoparticles were evaluated for the combined delivery of doxorubicin, sodium carbonate (Na_2_CO_3_) and liquid perfluorocarbon (PFC) for ultrasound-responsive antitumor treatment. The liquid PFC nanodroplets were evaporated by ultrasound to stimulate rapid Na_2_CO_3_ release. Na_2_CO_3_ acting as a neutralizing agent regulated the cellular proton pumps, and this resulted in inhibition of lactate acidosis and enhanced DOX release; thus increasing tumor growth inhibition.

As outlined by Shi et al. [252], the pH-sensitivity of drug delivery by nanomedicines can be contributed to by various mechanisms including protonation of biomolecules (drugs, peptides, and polymers) and degradation of pH-sensitive bonds. The study of nanoparticles composed of pH-sensitive copolymers selectively dissociating was investigated by Wang et al. [253]. They studied polymeric micelles composed of two types of polymers. One type was a Ce6-modified acid-responsive poly(ethylene glycol)-b-poly(2-(hexamethyleneimino) ethyl methacrylate) (PEG-b-PHMA) copolymers. The Chlorin e6 (Ce6) photosensitizer was grafted onto the acid-sensitive PHMA diblock. The second type of polymer was a iRGD peptide-modified polymeric prodrug of doxorubicin (iPDOX). The acid-responsive and peptide polymers were combined into one single nanoplatform (termed iPAPD). At physiological pH, the PEG-*b*-PHMA matrix was rigid, and this protected DOX prodrug from degradation and non-specific activity; while at the acidic tumor pH the PEG-*b*-PHMA chain was susceptible to degradation, releasing the DOX prodrug and restoring the Ce6 activity for real-time fluorescence imaging. The nanoparticles were evaluated in tumor spheroids and tumor-bearing animal models, and this showed effective tumor accumulation and increased tumor penetration due to the iRGD peptide. The P85 pluronic blocked the P-gp pumps and prevented DOX efflux; this further caused an elevated antitumor effect. The combination with PDT resulted in activation of Ce6 which induced ROS production, promoted DOX diffusion inside the tumor mass, and inhibited acquired drug resistance by altering the gene expression profile of the tumor cells. Another example of acid-responsive polymers was described by Liu et al. [254], who developed pH-sensitive amphiphilic block PCL-b-PEG copolymers for the co-delivery of paclitaxel (PTX) and acetazolamide (ACE). ACE is an inhibitor of carbonic anhydrase IX (CA IX) which is related to acidic tumor pH and MDR. The pH-responsiveness was attributed to the pH-cleavable hydrazine bond, which promoted the degradation of the polymeric shell and the release of ACE and PTX. In vivo administration of these dual-drug loaded nanoparticles resulted in successful tumor accumulation and tumor growth inhibition; this increased the survival rate of tumor-bearing mice. The ability of these nanoparticles to restore tumor acidity resulted in the enhanced effectiveness of paclitaxel. 

An alternative to polymers was provided by biomimetic nanoparticles composed of membranes originating from natural cells. Liu et al. [255] studied hybrid DSPE-PEOz pH-sensitive liposomes loaded with doxorubicin and incorporated into platelet membrane-coated nanoparticles (platesomes). Platesomes have a notable active tumor targeting behavior, since their membrane expresses several surface proteins including integrin α6, CD41, and CD62p; these specifically bind to the CD44 receptor of tumor cells. The hybrid nanoparticles expressed increased plasma half-life and elevated tumor accumulation, and this enables the selective release of DOX from the pH-sensitive liposomes in response to the acidity of lysosomes. Platesomes exhibited a significantly enhanced antitumor effect in 4T1 tumor-bearing BALB/c mice. Platelet membrane nanoparticles were also studied by Luo et al. [256] for their synergistic effect against tumor acidosis and hypoxia. Zeolitic imidazolate framework-8 nanoparticles (ZIF8) delivering doxorubicin, hemoglobin (Hb), and lactate oxidase (LOX) were further coated with platelet membrane to enhance passive targeting, increase circulation time, and lower toxicity in the biological environment. The nanoparticles synergistically combined the anticancer activity of DOX with the anti-hypoxia activity of oxygen-carrying hemoglobin and with the LOX catalytic activity in converting lactic acid to pyruvate and hydrogen peroxide. The evaluation in BALB/c tumor-bearing mice revealed an elevated tumor targeting effect of the nanoparticles; this degraded intracellularly to release Hb and LOX, thus inhibiting tumor hypoxia and acidity through oxygenation and lactate decomposition, respectively. The produced hydrogen peroxide resulted in oxidative stress of the tumor cells that, in combination with DOX, enhanced cellular apoptosis. The synergistic effects of the platelet nanoparticles resulted in suppressed tumor hypoxia, remodeling of tumor acidity, and inhibition of tumor growth. 

**Table 6 pharmaceutics-16-00179-t006:** pH-sensitive nanomedicines based on biomaterials targeting tumor acidosis.

Targeting Effects	Carrier Type	Therapeutic Agent	Characteristics	Ref.
pH-sensitive peptides	Chitosan nanoparticles/cRGD peptide	Raloxifene	Increased accumulation, enhanced antitumor effect inhibiting angiogenesis and migration	[226]
	Glycogen nanoparticles/hydrazine-based bond	Doxorubicin/β-galactose	Enhanced accumulation, inhibiting tumor growth	[227]
	PLGA–BSA particles ATRAM peptide	Doxorubicin/TPP	Enhanced mitochondria targeting, inhibited tumor volume and mass	[228]
	Hyaluronic acid nanogels E3/K3 peptides	Cytochrome C (CC)/saporin proteins	Inhibition of protein synthesis in the cytosol, efficient antitumor effect	[230]
Metals/Metal Oxides Chemo-Sensitivity	Cerium oxide–glycol chitosan nanoparticles	CXCR4 antagonist/Doxorubicin	Elevated internalization, increased ROS production at acidic pH, tumor size suppression, and reduced blood vessel leakage	[233]
	PEG–MnO_2_ nanoparticles	Doxorubicin/Ce6 PDT	Tumor oxygenation, inhibition of tumor growth, elevated antitumor immune responses	[234]
	MnO_2_-coated mesoporous silicon nanoparticles	Metformin/fluvastatin sodium	Induced intracellular acidosis promoting tumor cell death, suppressed tumor growth and metastasis	[235]
	Au nanorods/P(Glu-co-Lys) polypetides	Au nanorods	Enhanced accumulation in tumors periphery and hypoxic core	[236]
	Iron oxide SPIONs/cystamine-dextran	Doxorubicin	Increased pH-triggered internalization, inhibition of tumor volume	[237]
	Iron oxide SPIONs/PMAA-g-PEGMA	Canagliflozin/Radiotherapy	Accumulation in tumor tissue, inhibition of tumor growth	[238]
pH-sensitive Polymeric particles	Carboxyethyl chitosan–PEGDA hydrogels	Doxorubicin	Self-healing properties, antitumor effect	[243]
	Chitosan-PEG niosomes	Tamoxifen	Increased drug accumulation and antitumor efficacy	[244]
	Chitosan microparticles	-	Screening of tumor progression	[245]
	FA-PMgDP-PDPA-PDEMA particles	Doxorubicin/Galactose	Efficient internalization, increased toxicity, and apoptosis	[247]
	PCL-b-PAEP-TMA-Cya/DMA micelles	Doxorubicin	Enhanced internalization, inhibition of tumor growth	[248]
	Iron oxide-PDPA particles	PEG-polycamptothecin prodrug	Effective antitumor activities, effective antitumor activities	[249]
	Graphene quantum dots-PLGA-BSA particles	Doxorubicin	Sufficient internalization and in vitro toxicity	[250]
	PLGA particles	Doxorubicin/sodium carbonate/liquid perfluorocarbon	Tumor accumulating ability, and inhibited tumor growth	[251]
	PEG-b-PHMA particles	Doxorubicin-P85 prodrug/iRGD peptide/Ce6 PDT	Elevated antitumor effect and complete suppression of tumor growth	[253]
	PCL-PEG particles	Paclitaxel/Acetazolamide	Inhibitory effect on tumor growth, increasing the survival rate	[254]
	DSPE-PEOz liposomes in platelet membrane particles	Doxorubicin	Enhanced antitumor effect	[255]
	Zeolitic imidazolate framework-8 nanoparticles	Doxorubicin/hemoglobin/LOX	Tumor targeting effect, suppressed tumor hypoxia, remodeled tumor acidity and inhibited tumor growth	[256]

### 3.6. Tumor Immunotherapies

Another important area of nanomedicine research is tumor immunotherapy. Tumor immunotherapies may represent a vital solution in TME limitation and can be divided into immune checkpoint inhibitors (ICI), immune tumor vaccines, and chimeric antigen receptor-modified CAR T cells. The development of nanomedicines for cancer immunotherapy is largely based on the application of biomaterials such as polysaccharides (e.g., hyaluronic acid, alginic acid, chitosan, and dextran) and synthetic polymers including PEG, PLA, PLGA, PCL, and polypeptides [257]. The development of immunotherapeutic nanomedicines based on biomaterials has emerged as a strategy to improve therapeutic outcome by inducing tissue-specific immunomodulation through cell, antibody, and gene immune responses [258]. Opportunities to apply biomaterials in tumor immunotherapy and their combination with conventional therapies have been interestingly reviewed [259,260,261]. Among immunotherapies, immune checkpoint inhibitors present promising therapies that mainly target cytotoxic T lymphocyte-associated molecule-4 (CTLA-4), programmed cell death receptor-1 (PD-1), and programmed cell death receptor-1 ligand (PD-L1). Immune checkpoints are proteins on the surface of immune T cells that recognize and bind to partner proteins on tumor cells, and promote intracellular inhibitory signals and immunosuppressive enzymes; this suppresses host immune T cell attack against tumor cells. Hu et al. [262] developed amphiphilic nanoparticles with a core from hydrophobic ROS-sensitive poly(thioketal phosphoester) and a shell from hydrophilic lecithin/DSPE-PEG, encapsulating doxorubicin, Ce6 photosensitizer, and anti-PD-L1 antibody for ICI. The nanoparticles were evaluated in 4T1 tumor-bearing mice. Rapid degradation of the ROS-responsive core triggered DOX release and, in combination with laser irradiation, promoted effective PDT activity due to the Ce6 component of the nanoparticles. The combinational effect with the anti-PD-L1 antibody ICI promoted the maturation of DCs, and this significantly suppressed the growth of primary tumors and effectively inhibited distant tumor growth. Another type of immunotherapy is the delivery of immunostimulatory agents that directly activate immune T cells by binding of agonistic antibodies on surface receptors such as CD40, OX40 (CD134), and CD137; this promotes downstream signaling pathways for T cell-mediated antitumor activity [263,264,265,266].

The application of biomaterials in cancer vaccines has made significant progress due to the advantageous effect on enhancing the safety profile and stimulating antigen-specific T cell response. Immunotherapy vaccines aim to activate the immune system attack against tumor cells by downregulating the immune tolerance to tumor antigens, through the combined delivery of immunostimulatory agents to activate host immune cells and immunogenic epitopes of specific tumor antigens [263,264]. Τhe effective delivery of the immunotherapy vaccines to dendritic cells is crucial, because DCs are the main antigen presenting immune cells. Rosalia et al. [264], reported that PLGA nanoparticles coated with agonistic aCD40-mAb and encapsulating tumor associated Ag protein successfully primed CD8^+^ T cells; thus, the survival of tumor-bearing mice was prolonged. In another study, Wang et al. [265] investigated amphiphilic pH-sensitive galactosyl dextran-retinal (GDR) nanogels with a pH-sensitive hydrazone bond for dextran conjugation with all-trans retinal (a metabolite of vitamin A). The nanogels were galactosylated to acquire DC-targeting ability and were effective vaccine delivery systems due to their ability to successfully amplify major histocompatibility class I, MHC I, antigen expression in DCs, and induced effective antitumor immune responses.

The most recent immunotherapies are CAR T cell and CAR T cell receptor therapies. CAR T cells are engineered immune cells originating from the peripheral blood mononuclear cells of the patient’s blood that are harvested and stimulated to become T cells with specific DNA encoding to recognize certain tumor antigens. There are two FDA-approved CAR T cell therapies for blood cancer; however, their application in solid tumors remains challenging. Nevertheless, CAR T cell therapies against solid tumors are being studied ongoing clinical trials [263,264]. Biomaterials are promising candidates for CAR T cell modification improving the therapeutic efficacy and immune-editing processes [266]. Moreover, nano-biomaterials can improve the effect of immunotherapies, and trigger enhanced therapeutic outcomes and regulation of immune cells by overcoming the barriers of the cold tumor immune microenvironment [267].

## 4. Discussion

Nanomedicine is a widely applied research field for responsive therapies that target the TME of solid tumors. The dynamic features of TME represent the greatest barrier to effective delivery of nanomedicines and influence the development of acquired and multidrug resistance by tumor cells. The high heterogeneity of solid tumor microenvironments promotes the evolution of cancer stem cells, and their invasion and metastasis. Great effort has been devoted in exploiting the specific features of TME by nanomedicine and functional biomaterials. Such efforts have resulted in nanomedicines for the degradation of ECM components, the site-specific targeting of cancer-associated cells, the disruption of glycolysis, the increase of oxygenation, and the pH-responsiveness [268,269,270]. Still, conventional cancer therapies remain the main cancer treatment, while TME-responsive nanomedicines that are FDA-approved are mostly applied in combination therapies but not as monotherapies. In recent years there has been great progress in overcoming TME obstacles either by combining nanomedicines with multiple endogenous responses in a single system or by regulating tumor responses with the effect of an external stimuli such as PDT, SDT, and PTT; this leads to an augmented therapeutic outcome. Polymer nanoparticles based on bioresponsive biomaterials have been widely studied. For example, Chen R. et al. [271] effectively applied polymeric PEG-b-PBS nanoparticles in dual responsive targeting and controlled release of doxorubicin. Tumor accumulation of the PEG-b-PBS nanoparticles was accomplished due to passive targeting (EPR phenomenon) and at the tumor area, degradation of the phenylboronic ester was induced by the effect of extracellular ROS. Following nanoparticles cellular uptake, the effect of elevated GSH levels in the cytoplasm and increased thiol levels in lysosomes caused the lysis of the disulfide bonds on the phenylboronic ester, and this triggers DOX release. The dual redox-responsiveness resulted in enhanced tumor inhibitory effect and fewer side effects. In another study by Chen X. et al. [272], DSPE-mPEG liposomes delivering S-nitroso-N-acetylpenicillamine (SNAP), a nitric oxide (NO) donor, and gemcitabine were evaluated for the combinational treatment of pancreatic ductal adenocarcinoma (PDAC). The inhibition of the dense PDAC stroma was achieved by suppressing TGF-β1 expression levels. The liposomes were effectively accumulated at the tumor site where they effectively degraded the dense stroma for increased gemcitabine penetration into the tumor both in subcutaneous and in orthotopic tumor-bearing mice. Dabaghi et al. [273] studied chitosan-coated magnetic nanoparticles functionalized with 5FU for combinational magnetic hyperthermia and targeted antitumor activity. Evaluation of HT-29 tumor-bearing mice under the effect of an alternating magnetic field showed an effective antitumor activity with regulation of ECM protein levels. From computation analysis, elevated DNA damage was effectively predicted, as were increased cellular stress and modifications in receptor signaling and immune responses. The observations signified the need to repeat therapeutic cycles of chemotherapy and hyperthermia. Paholak et al. [274] studied local hyperthermia by highly crystallized iron oxide nanoparticles to facilitate PTT and promote effective inhibition of metastasis in NOD/SCID mice bearing triple-negative breast cancer. Local hyperthermia effectively suppressed breast cancer stem cells (BCSCs) and considerably inhibited metastasis to the lung and lymph nodes. This signifies the importance of PTT in tumor therapeutic applications. Tan et al. [275] studied the combination of chemotherapy and PDT using biomimetic lipoprotein particles (BL-RD) composed of phospholipids and apolipoprotein A1 mimetic peptide (PK22). The particles were loaded with mertansine and the photodynamic agent DiIC18(5) (DiD) and further conjugated with CRGDfK targeting peptide. These nanoparticles were tetsed in 4T1 tumor-bearing mice and they exhibited tumor targeting specificity, deep tissue penetration, and internalization by stromal cells such as TAM, CAF, and EC. This is due to the targeting peptide and the biomimetic character. Mertansine presented an effective cytotoxic effect; it induced mitotic arrest and promoted tumor cell death. The combined activity of DiD with PDT laser resulted in the efficient production of ROS and further promoted inhibition of tumor growth. An interesting review by Kola et al. [276] presents the combination of chemotherapy with PTT, PDT, and SDT in theranostic nanomedicine applications for effective treatment of breast cancer. This outlines the mechanisms of action and the therapeutic efforts to overcome breast cancer stem cells. 

This great effort resulted in an exponential increase of the pre-clinical studies of nanomedicines. However, many trials fail to achieve their goal; nearly 6% of nanomedicines pass from phase I to clinical approval [277]. Despite the advancements in cancer nanomedicines and the great efforts that have been made in the field of combinational therapeutics, there are still unresolved challenges for their successful clinical translation [278]. Nanomedicines with effective biological and physicochemical characteristics in biocompatibility, size, shape, and surface chemistry provide promising results in the pre-clinical stage. A great obstacle that remains in the direction of clinical translation of cancer nanomedicines is the effective accumulation at tumor side. The effects of design parameters, materials used, and targeting processes are critical for tumor accumulation of the nanomedicines. Moreover, deep penetration and stimuli-responsive drug release should be achieved to overcome the heterogenic vasculature and thereby increase the therapeutic index of the therapeutics. The toxicological characteristics of the degradation byproducts of the nanomedicines is another challenge that novel nanomedicines try to overcome by the use of biocompatible materials and biomacromolecules. The high genetic heterogeneity of solid tumor microenvironments remains a major obstacle that has been considerably researched by personalized nanomedicine and immunotherapies [277,278]. 

## 5. Conclusions

Responsive biomaterials are broadly researched for the development of nanomedicines that target the TME of solid tumors; they have attracted considerable interest in both fundamental research studies and clinical applications. Natural and synthetic biomaterials are essentially investigated in varied nanomedicine due to their ability of functionalization in order to obtain internal and external stimuli responsiveness and to carry therapeutic and diagnostic agents. Their TME-targeting applications have spread in nanomedicines used for the heterogenic vasculature, tumor stroma extracellular matrix and CAFs, hypoxia, and acidosis. Nanomedicines based on biomaterials have great prospects in terms of immune reprogramming and cancer vaccines for novel therapeutic approaches. The effect of responsive biomaterials in combinational therapy is significant, and this has triggered intense research interest in the evaluation of signaling pathways and mechanisms associated with multidrug resistance and immunosuppression. Moreover, the extensive development of nanomedicines over the past two decades promoted immunotherapies that focus on triggering the patients’ immune system ability to recognize and attack cancer cells. In this review, the theory of targeting nanomedicines that are based on biomaterials is presented for the treatment of solid tumors through specific responses in the dense and heterogeneous tumor microenvironment.

## 6. Future Directions

The application of nanomedicine in cancer therapy has generated great advancements especially in the development of novel strategies, including combinational therapies and immunotherapies. The combination of responsive nanomedicines for overcoming or exploiting TME of solid tumors and external stimuli including PDT/PPT, hyperthermia, and ultrasound, has resulted in the application of high-intensity focused ultrasound (HIFU) in prostate treatment. This is an FDA-approved, minimally invasive procedure. Moreover, cancer immunotherapies have received enormous attention, with two CAR T cells therapies being approved by the FDA in 2017 for advanced/resistant lymphoma and acute lymphoblastic leukemia. However, great concerns still remain around the safety of CAR T cells for patients. Safety and efficacy concerns remain limited to the translation of nanomedicine therapies from the laboratory and preclinical stage to clinical application. The main obstacles for the clinical translation of cancer-responsive nanomedicines are the complicated manufacturing processes that pose significant difficulties in large-scale production, the biological interactions of nanomedicines with the protein environment in vivo, the potential toxicological reactions of the nanomedicines byproducts, and above all, the high genetic heterogeneity of tumor microenvironment among patients. Further research in these directions and especially in the heterogeneity of TME is needed to bring into the clinic responsive TME-targeting nanomedicines for personalized cancer therapy. Adopting a more “systems approach” integrating the cross-talking and interactions of all TME components, in our opinion, could be a more meaningful way to exploit TME features and alleviate TME obstacles in our efforts to develop efficacious anticancer medicines and nanomedicines.

## Data Availability

No new data were used.

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
