# Peer review of "Biomaterial-Based Responsive Nanomedicines for Targeting Solid Tumor Microenvironments"

_pharmaceutics, 2024, doi:10.3390/pharmaceutics16020179_

Round 1

Reviewer 1 Report

Comments and Suggestions for Authors

The article is very extensive in terms of covering all aspects but then loses a little in specificity as it addresses all strategies in the treatment of solid tumors, not focusing on "Targeting Solid Tumor Microenvironment" as described in the title.

The Scheme 1. Needs to be improved in order the legends are readable. I suggest separating A, B and C. The description of C is missing.

The article needs major improvements in the figures and diagrams it presents because they are not very readable.

Furthermore, some aspects need to be updated as references are made to products that, despite having been approved, have since been withdrawn from the market. Therefore, I suggest you update what is still on the market.

Author Response

Reviewer 1

The article is very extensive in terms of covering all aspects but then loses a little in specificity as it addresses all strategies in the treatment of solid tumors, not focusing on "Targeting Solid Tumor Microenvironment" as described in the title.

The Scheme 1. Needs to be improved in order the legends are readable. I suggest separating A, B and C. The description of C is missing.

The article needs major improvements in the figures and diagrams it presents because they are not very readable.

Answer

All schemes of the review paper were redesigned in order to become simple and readable. Some schemes were changed in order to become easier for the reader to understand the targeting processes presented

Furthermore, some aspects need to be updated as references are made to products that, despite having been approved, have since been withdrawn from the market. Therefore, I suggest you update what is still on the market.

Answer

Table 1, page 5 has been changed according to Reviewer’s recommendations. Specifically, dates for discontinued products were added and links were added for EMA or FDA approval pages. In the main paper the references to products is in terms of the developed systems and the potential nanomedicines can have in clinical practice. 

Reviewer 2 Report

Comments and Suggestions for Authors

In this review, the author has discussed natural and synthetic biomaterials applied to a variety of nanomedicines targeting the solid tumor microenvironment, including the heterogenic vasculature, tumor stroma extracellular matrix and cancer associated fibroblasts (CAFs), hypoxia, and acidosis. This review elaborates various targeting strategies according to the heterogenic microenvironment of tumor and the tables listed are convincing.

Some issues need to be addressed as follows.

1.      The text needs to be carefully edited. The authors need to pay attention to English gramma and sentence structure, such as “suppression of cancer stem cells” in the abstract. Besides, most of the sentences are too long and difficult to understand.

2.      In the last paragraph of the introduction, the author has listed several applications of biomaterial-based nanomedicine in targeting TME, among which the crosslinking efficacy and co-delivery function of various diagnostic or therapeutic agents by biomaterial should be effectively summarized into one point.

3.      This review emphasizes on the targeting theory of nanomedicine. Please give one or more examples including the schematic diagrams for each targeting strategy to present more clearly.

4.      The keyword throughout the text is “targeting nanomedicines” or “targeting formulations” which is not quite match the title -- “responsive nanomedicines”. Furthermore, they are two concepts at different levels.

5.      The authors are suggested to summarize the main points and highlight the innovative contributions of this work.

6.      For the future directions in nanomedicines for targeting solid TME, there has been lots of research on the combination of TME targeting and immunotherapies, as well as other combination therapy, such as chemotherapy, chemodynamic therapy, photodynamic, sonodynamic and photothermal therapy. Please summarize the challenges of clinical translation of nanomedicines and provide future perspectives of the combination therapy of TME targeting.

Author Response

Reviewer 2

In this review, the author has discussed natural and synthetic biomaterials applied to a variety of nanomedicines targeting the solid tumor microenvironment, including the heterogenic vasculature, tumor stroma extracellular matrix and cancer associated fibroblasts (CAFs), hypoxia, and acidosis. This review elaborates various targeting strategies according to the heterogenic microenvironment of tumor and the tables listed are convincing.

Some issues need to be addressed as follows.

  1. The text needs to be carefully edited. The authors need to pay attention to English gramma and sentence structure, such as “suppression of cancer stem cells” in the abstract. Besides, most of the sentences are too long and difficult to understand.

Answer

Editing of the whole manuscript has been carefully performed in accordance to Reviewer’s suggestions. English grammar and syntactic corrections have been corrected throughout the review paper. The abstract with the mentioned sentence “suppression of cancer stem cells” was corrected. All changes in the abstract are in yellow highlight. Moreover, the long sentences were shortened in order to be understood.

  1. In the last paragraph of the introduction, the author has listed several applications of biomaterial-based nanomedicine in targeting TME, among which the crosslinking efficacy and co-delivery function of various diagnostic or therapeutic agents by biomaterial should be effectively summarized into one point.

Answer

The last paragraph of the introduction was corrected and the application of biomaterials was summarized. The changes are in yellow highlight.

  1. This review emphasizes on the targeting theory of nanomedicine. Please give one or more examples including the schematic diagrams for each targeting strategy to present more clearly.

Answer

All the schemes of the review paper were redesigned in order to become simple and readable. Some schemes were changed in order to be easier for the reader to understand the targeting processes presented.

  1. The keyword throughout the text is “targeting nanomedicines” or “targeting formulations” which is not quite match the title -- “responsive nanomedicines”. Furthermore, they are two concepts at different levels.

Answer

The parts that connected with such misunderstandings were corrected in order to be in line with the title of the review, as the Reviewer suggested. The corrections are in yellow highlight (pages 6, 12, 13, 17, 29 table 6, 30, and 32)

  1. The authors are suggested to summarize the main points and highlight the innovative contributions of this work.

Answer

The Conclusions part (pages 36-37) was corrected in accordance to the reviewers’ suggestions to follow the presented work and summarize the critical points.

  1. For the future directions in nanomedicines for targeting solid TME, there has been lots of research on the combination of TME targeting and immunotherapies, as well as other combination therapy, such as chemotherapy, chemodynamic therapy, photodynamic, sonodynamic and photothermal therapy. Please summarize the challenges of clinical translation of nanomedicines and provide future perspectives of the combination therapy of TME targeting.

Answer

The part of Future directions was changed in accordance to Reviewer’s recommendation and is in yellow highlight. Moreover, an extra paragraph was added in the discussion part summarizing the importance of clinical translation (pages 36,37)

Reviewer 3 Report

Comments and Suggestions for Authors

In this review, the authors have focused on the targeting mechanisms for solid TME by nanomedicines based on the application of natural and synthetic biomaterials. In addition, the authors have described critical formulations that have been considered for the design of stimuli-responsive nanomedicines for TME. Overall, it's timely and well written. However, there are several minor issues should be addressed before acceptance.

1. The tumors also have the immunosuppressive TME characteristics. Thus, the author should also discuss in detail the strategies of delivery of nanomedicines to reverse the tumor immunosuppressive microenvironment for tumor treatment.

2. Several responsive nanomedicine strategies for improving hypoxia in tumor treatment have not received attention, such as J Control Release. 2022, 345:755-769; Smart Materials in Medicine 2023, 4:286-293.

3. The text font in Scheme 2 is not clear enough.  

Author Response

Reviewer 3

In this review, the authors have focused on the targeting mechanisms for solid TME by nanomedicines based on the application of natural and synthetic biomaterials. In addition, the authors have described critical formulations that have been considered for the design of stimuli-responsive nanomedicines for TME. Overall, it's timely and well written. However, there are several minor issues should be addressed before acceptance.

  1. The tumors also have the immunosuppressive TME characteristics. Thus, the author should also discuss in detail the strategies of delivery of nanomedicines to reverse the tumor immunosuppressive microenvironment for tumor treatment.

Answer

An extra separate part (3.6 Tumor Immunotherapies, pages 34-35) was added, where the importance of these therapeutic approaches is presented. The therapeutic approaches of CAR T cells, immune reprogramming and immune vaccines are presented, in terms of nanomedicine formulations that are based on biomaterials, as in the rest of this review. The changes are in yellow highlight.

  1. Several responsive nanomedicine strategies for improving hypoxia in tumor treatment have not received attention, such as J Control Release. 2022, 345:755-769; Smart Materials inMedicine 2023, 4:286-293.

Answer

The nanomedicine strategies suggested have been added in page 26, 27, references 200 and 201, respectively. The changes are in yellow highlight and references have been changed accordingly.

  1. The text font in Scheme 2 is not clear enough.  

Answer

All schemes and diagrams were improved in order to become simple and readable.

Round 2

Reviewer 1 Report

Comments and Suggestions for Authors

The authors consider all observations made to the manuscript improving it. Thus, in my opinion, the article may be considered for publication.

Reviewer 2 Report

Comments and Suggestions for Authors

The authors have revised the manuscript according to the revewer's comments.